# Severely Ill COVID-19 Patients May Exhibit Hypercoagulability Despite Escalated Anticoagulation

**DOI:** 10.3390/jcm14061966

**Published:** 2025-03-14

**Authors:** Soslan Shakhidzhanov, Anna Filippova, Elizaveta Bovt, Andrew Gubkin, Gennady Sukhikh, Sergey Tsarenko, Ilya Spiridonov, Denis Protsenko, Dmitriy Zateyshchikov, Elena Vasilieva, Anna Kalinskaya, Oleg Dukhin, Galina Novichkova, Sergey Karamzin, Ilya Serebriyskiy, Elena Lipets, Daria Kopnenkova, Daria Morozova, Evgeniya Melnikova, Alexander Rumyantsev, Fazoil Ataullakhanov

**Affiliations:** 1Dmitriy Rogachev National Medical Research Center of Pediatric Hematology, Oncology, and Immunology, 117997 Moscow, Russia; ae.zadorozhnaya@physics.msu.ru (A.F.); elizaveta.bovt@dgoi.ru (E.B.); novichkova.galina@dgoi.ru (G.N.); morozova.daria@ctppcp.ru (D.M.); alexander.rumyantsev@dgoi.ru (A.R.); 2Center for Theoretical Problems of Physicochemical Pharmacology, 109029 Moscow, Russia; spiridonov.ilya@ctppcp.ru (I.S.); karamzin.sergey@ctppcp.ru (S.K.); serebriyskiy.ilya@ctppcp.ru (I.S.); lipets.elena@ctppcp.ru (E.L.); melnikova.evgeniya@ctppcp.ru (E.M.); 3Central Clinical Hospital No. 2 Named After N.A.Semashko “RZD-Medicine”, 121359 Moscow, Russia; gubkinav@gmail.com; 4National Medical Research Center for Obstetrics, Gynecology and Perinatology Named After Academician V.I.Kulakov, 117997 Moscow, Russia; g_sukhikh@oparina4.ru; 5City Clinical Hospital No. 52 of Moscow Health Care Department, 123182 Moscow, Russia; s9637501492@yandex.ru; 6Moscow Multiprofile Clinical Center “Kommunarka” of Moscow Healthcare Department, 142770 Moscow, Russia; drprotsenko@me.com (D.P.); primadaria95@gmail.ru (D.K.); 7City Clinical Hospital No. 51 of Moscow Health Care Department, 121309 Moscow, Russia; dz@bk.ru; 8City Clinical Hospital No. 23 of Moscow Health Care Department, 109004 Moscow, Russia; vasilievahelena@gmail.com (E.V.); kalinskaya.anna@gmail.com (A.K.); dukhinoa@zdrav.mos.ru (O.D.)

**Keywords:** LMWH, UFH, thromboprophylaxis, Thrombodynamics

## Abstract

**Introduction:** Severely ill COVID-19 patients receiving prophylactic-dose anticoagulation exhibit high rates of thrombosis and mortality. The escalation of anticoagulation also does not reduce mortality and has an uncertain impact on thrombosis rates. The reasons why escalated doses fail to outperform prophylactic doses in reducing risks of thrombosis and death in severely ill COVID-19 patients remain unclear. We hypothesized that escalated anticoagulation would not effectively prevent hypercoagulability and, consequently, would not reduce the risk of thrombosis and death in some severely ill patients. **Methods:** We conducted a prospective multicenter study that enrolled 3860 COVID-19 patients, including 1654 severely ill. They received different doses of low-molecular-weight or unfractionated heparin, and their blood coagulation was monitored with activated partial thromboplastin time, D-dimer, and Thrombodynamics. A primary outcome was hypercoagulability detected by Thrombodynamics. Blood samples were collected at the trough level of anticoagulation. **Results:** We found that escalated anticoagulation did not prevent hypercoagulability in 28.3% of severely ill patients at the trough level of the pharmacological activity. Severely ill patients with such hypercoagulability had higher levels of inflammation markers and better creatinine clearance compared to severely ill patients without it. Hypercoagulability detected by Thrombodynamics was associated with a 1.68-fold higher hazard rate for death and a 3.19-fold higher hazard rate for thrombosis. Elevated D-dimer levels were also associated with higher hazard rates for thrombosis and death, while shortened APTTs were not. The simultaneous use of Thrombodynamics and D-dimer data enhanced the accuracy for predicting thrombotic events and fatal outcomes in severely ill patients. **Conclusions:** Thrombodynamics reliably detects hypercoagulability in COVID-19 patients and can be used in conjunction with D-dimer to assess the risk of thrombosis and death in severely ill patients. The pharmacological effect of LMWH at the trough level might be too low to prevent thrombosis in some severely ill patients with severe inflammation and better creatinine clearance, even if escalated doses are used.

## 1. Introduction

Patients with COVID-19 are at substantially increased risk for both venous and arterial thrombotic complications, as well as a higher risk of fatal outcomes [1]. It is now well established that SARS-CoV-2 infection profoundly disrupts hemostatic balance through a complex interplay of multiple concurrent mechanisms. These include platelet hyperreactivity [2], elevated levels of von Willebrand factor [3], the active formation of neutrophil extracellular traps [4], which in turn triggers platelet activation [5], and blood coagulation cascade hypercoagulability driven by increased circulating extracellular vesicles containing tissue factor [6,7].

Addressing each of these factors requires distinct approaches. In particular, one of the most effective methods for managing COVID-19-induced thrombophilia has been the use of low-molecular-weight heparins (LMWHs) [8,9,10]. Consequently, international hematology societies, including the International Society on Thrombosis and Haemostasis (ISTH), have recommended the routine administration of LMWHs to all inpatients with COVID-19 [11,12]. Nevertheless, even with the administration of LMWHs, the risks of death and thrombotic complications among severely ill patients remain substantial: the frequency of thrombotic complications can reach up to 24.1% [13], while the mortality rate reaches 17.1% [14]. It was hypothesized that these risks would be significantly reduced by using escalated LMWH doses in severely ill patients [15,16].

However, randomized clinical trials have shown that dose escalation does not improve survival in severely ill patients [17,18,19,20,21,22,23,24,25]. Additionally, the effect of dose escalation on the rate of thrombotic events is unclear; some studies have shown a reduction in the thrombosis rate [17,19,24], while others do not support this [18,20,21,22,23,25]. It is unexpected, since severely ill patients frequently experience thrombotic events and have near-normal antithrombin III levels [26,27]. With uncertain benefits, escalated doses are associated with a higher rate of non-fatal bleeding in severely ill patients [17,19]. So, current guidelines recommend prophylactic doses for severely ill patients [28]. In contrast, mildly ill patients benefit from escalated doses, as they slightly improve their survival and do not increase the bleeding rates [29,30].

Hence, LMWH escalation is not sufficient to fully mitigate the risks of thrombosis and lethal outcome in severely ill COVID-19 patients. These unmitigated risks may stem from several concurrent factors. This may be due to platelet hyperreactivity, which leads to thrombosis [31,32,33,34,35]. However, some studies instead indicate that platelet activity is suppressed in severely ill patients with coronavirus infection [36,37], and clinical trials of antiplatelet therapy have shown that it is not effective in such patients [17,38,39]. Another factor may be that thrombus formation in severely ill patients is driven by inflammation [40], and increasing the dose of anticoagulants does not reduce the tendency to pathological thrombus formation.

However, another potential concurrent factor is that escalated LMWHs still do not effectively prevent hypercoagulability in severely ill patients, especially at the trough level of the pharmacological activity, and thus do not reduce the risks of thrombosis and mortality. To date, this factor remains poorly studied.

One of the methods for assessing blood coagulation is “Thrombodynamics”, a global assay of hemostasis that considers spatial heterogeneity of blood clotting and imitates blood coagulation in vivo [41]. It has been used to assess the risk of thrombotic events in septic patients, monitor heparin treatment in patients at high risk of venous thromboembolism (VTE), and correct episodic hypercoagulability in a limited cohort of COVID-19 patients [36,42,43]. Thrombodynamics assay description and clinical application results are given in Appendix A.

The objective of our study was to determine whether escalated (intermediate and therapeutic) LMWH prevented hypercoagulability in COVID-19 patients at the trough level of the pharmacological activity. We also studied whether hypercoagulability was associated with higher risks of thrombosis and death. We conducted a prospective multicenter study that enrolled 3860 patients, including 1654 severely ill. They received prophylactic, intermediate, and therapeutic doses of “heparins” (LMWHs) or unfractionated heparin (UFH). Coagulation was assessed using activated partial thromboplastin time (APTT), D-dimer, and the Thrombodynamics assay.

## 2. Methods

### 2.1. Study Design

The prospective study was conducted from July 2020 to September 2021 in seven hospitals in Moscow, Russia. Patients were admitted from home via emergency medical services (ambulance) or transferred from other hospitals (Figure 1). The inclusion criteria were age between 18 and 100 years, a requirement for hospitalization, consent from the patient or their representative to participate in the study, a preliminary diagnosis of COVID-19, and the patient’s condition being classified as mild or severe. The exclusion criteria were the refusal of the patient or their representative to participate in the study, COVID-19 diagnosis was not confirmed through PCR, and pregnancy. Patients were followed up until their in-hospital death or discharge. Their blood coagulation was assessed using APTT, D-dimer, and a clot growth rate in the Thrombodynamics assay (TDX-V). During the study, parameters of biochemistry and complete blood count were also assessed.

Patients were classified as severely ill if they required for admission to the intensive care unit (ICU) during hospitalization. Patients who did not require for admission to ICU during hospitalization were classified as mildly ill. Patients required admission to ICU if two of the following criteria were met: (1) consciousness alterations, (2) SpO_2_ < 92% while receiving respiratory support, and (3) respiratory rate > 35/min.

Blood samples were collected from the medial cubital vein or venous catheter. Samples for blood coagulation assays were collected upon admission to the hospital, before anticoagulation treatment was started. Subsequent samples were collected at the trough level of anticoagulation, at least every three days (in the morning while fasting), or daily in cases of severe illness, unless contraindications were present. For UFH infusion, where no clear trough level exists, samples were collected at least 6 h after the bolus or after adjustment of the infusion rate.

Blood for coagulation assays was drawn into 4.5 mL plastic test tubes containing 3.2% sodium citrate (UNIVAC, Moscow, Russia). The initial 1–2 mL of blood was discarded into an empty tube.

Patients were regularly screened for thromboembolic complications, or screenings were performed on an emergency basis if a thrombotic complication was suspected. Thrombotic events were identified by instrumental diagnosticians and radiologists and confirmed by the treating physicians of the patients. Computed tomography with or without angiography, ultrasound imaging results, or laparoscopy were used for identification. Three members of our research group reviewed these cases to determine whether they were acute events that occurred after the patients’ admission to the hospital.

Electronic medical records were used to collect demographic data, treatments, assay results and outcomes. Data extraction was performed with documentation of the retrieval process. Data were collected from August 2020 to October 2021.

All hospitalized patients were treated according to the temporary Russian Federation National Guidance for Treating COVID-19, using versions 6 to 12, which were valid during the study period (version 12 translated using Google Translate© is attached—Appendix A). It states that all hospitalized COVID-19 patients should receive LMWHs or fondaparinux sodium at least in prophylactic doses. UFH could be used as an alternative. Prophylaxis should be maintained during the whole hospitalization period. Patients with severe COVID-19 symptoms, additional thrombotic risk factors, or markedly elevated D-dimer levels were candidates for escalation to intermediate or therapeutic doses of LMWH/UFH. At the same time, randomized controlled trials have not demonstrated that routine escalation to intermediate or therapeutic doses of anticoagulants improved clinical outcomes in critically ill patients in the ICU. Therapeutic dosages of LMWH/UFH/fondaparinux sodium should have been indicated for a minimum of three months in patients with suspected/confirmed pulmonary embolism (PE) or deep vein thrombosis (DVT).

The following heparins were used during the study: prophylactic (UFH—≤5000 IU TID s/c; dalteparin 5000 IU/day s/c; enoxaparin—4000 IU/day s/c; nadroparin—2850–3800 IU/day if <70 kg or 3800–5700 IU/day if ≥70 kg s/c), intermediate (UFH >5000 IU TID s/c; dalteparin 5000 IU/BID s/c; enoxaparin—4000 IU/BID s/c; nadroparin—3800–5700 IU/BID s/c) and therapeutic (UFH i/v infusion; dalteparin 100 IU/kg/BID s/c; enoxaparin—100 IU/kg/BID s/c; nadroparin: 86–100 IU/kg/BID s/c).

The infusion of unfractionated heparin (UFH) began with a bolus dose of 80 units per kilogram of body weight followed by a continuous infusion of 18 units per hour per kilogram. The infusion rate was then adjusted based on APTT values (target range was 45–70 s; if <35 s, 80 U/kg re-bolus and +4 U/kg/h; if 35–45 s, 40 U/kg re-bolus and +2 U/kg/h; if 71–90 s, −2 U/kg/h; if >90 s, hold for one hour and then −3 U/kg/h). APTT was measured 3–4 times daily in patients receiving UFH infusion.

Therapeutic or intermediate doses of anticoagulants were recommended for patients in the intensive care unit (ICU), while prophylactic or intermediate doses were suggested for patients in the medical ward. Also, therapeutic doses of heparins were recommended for patients with atrial fibrillation, mechanical heart valves, or confirmed VTE/PE. Anticoagulation de-escalation was recommended for patients with kidney injury. Additionally, anticoagulant therapy could be temporarily lowered or discontinued in cases of bleeding, intubation, or surgical intervention. We did not analyze coagulability if anticoagulation was discontinued. A flowchart diagram of the algorithm for the dosing of LMWHs and UFH based on the patient’s history and clinical condition is given in Figure 2.

The dosing of heparins based on patients’ history and clinical condition could introduce bias into our study, e.g., it is possible that heparins’ effectiveness varied among severely ill patients with differing risk factors. We analyzed whether the group of severely ill patients was homogeneous in terms of Thrombodynamics clot growth rate—specifically, whether the obtained TDX-V distributions depended on the presence or absence of certain risk factors in patients. The results are described in the “Results” section (Appendix A).

### 2.2. Thrombodynamics Assay

Thrombodynamics is used to assess blood coagulation and monitor anticoagulation. It was conducted with a diagnostic system “Thrombodynamics Analyzer T2-F” (LLC HemaCore, Moscow, Russia) and kits provided by the manufacturer.

Briefly, this assay measures clot growth velocity (TDX-V) in platelet-free plasma derived from a site with immobilized tissue factor (Bedford, MA, USA). A TDX-V >29 μm/min suggests hypercoagulability, while a TDX-V <20 μm/min suggests hypocoagulability [41].

An assay was performed no longer than one hour after blood collection. Blood samples were processed by centrifugation at 1600× *g* for 15 min to obtain platelet-poor plasma. Then, it was repeatedly processed by centrifugation at 10,000× *g* for 5 min to obtain platelet-free plasma, which was used for the assay. One half-milliliter of platelet-free plasma was frozen and kept at −80 °C. Corn trypsin inhibitor (LLC Hemacore, Moscow, Russia) was added into the remaining volume of platelet-free plasma, then it was shaken. The plasma was then heated to 37 °C for 5 min, followed by the addition of calcium acetate (LLC Hemacore, Moscow, Russia); the plasma was then shaken, after which the analysis was immediately performed at 37 °C.

### 2.3. Other Assays

Collected samples of platelet-free plasma were thawed at room temperature and used for antithrombin III level measurements, which were performed according to the Liquid Antithrombin HemosIL (Instrumentation Laboratory, Bedford, MA, USA) kit protocol using an automated coagulometer ACL TOP 500 (Instrumentation Laboratory, MA, USA).

Other assays were performed in the hospitals by the staff of their clinical diagnostic laboratories using automated analyzers. The complete blood count was performed using hematological analyzers Sysmex XS-1000i (Sysmex Corporation, Kobe, Japan) and Horiba Pentra XLR (Horiba ABX SAS, Montpellier, France). The measurement of ferritin, C-reactive protein, creatinine, glucose, alanine and aspartate aminotransferases, lactate dehydrogenase, and bilirubin levels was carried out with automated biochemical analyzers Abbott Architect c4000 (Abbott, Lake Forest, USA) and ROCHE Cobas C311 (Roche Diagnostics, Rotkreuz, Switzerland) using the corresponding reagents. The analyses of APTT, prothrombin time (PT), D-dimer, and fibrinogen were performed with automated coagulometers Sysmex CS 2100i (Sysmex Corporation, Kobe, Japan), ACL TOP 500 (Instrumentation Laboratory, MA, USA), and ACL TOP 300 (Instrumentation Laboratory, MA, USA).

### 2.4. Materials

Pathromtin SL (Siemens Healthcare Diagnostics Products GmbH, Duisburg, Germany), SynthASil (Instrumentation Laboratory, Bedford, MA, USA), Thromborel S (Siemens Healthcare Diagnostics Products GmbH, Duisburg, Germany), ThromboPlastin (Instrumentation Laboratory, Bedford, MA, USA), INR Validate (Instrumentation Laboratory, Bedford, MA, USA), INNOVANCE D-Dimer (Siemens Healthcare Diagnostics Products GmbH, Duisburg, Germany), D-dimer HS (Instrumentation Laboratory, Bedford, MA, USA), Dade Thrombin (Siemens Healthcare Diagnostics Products GmbH, Duisburg, Germany), QFA Fibrinogen (Instrumentation Laboratory, Bedford, MA, USA), Liquid Antithrombin HemosIL (Instrumentation Laboratory, Bedford, MA, USA).

### 2.5. Outcomes

The primary outcome was hypercoagulability detected by Thrombodynamics clot growth rate in severely ill and mildly ill patients (TDX-V > 29 μm/min). The exposure was heparins dosage received by a patient. The collected data were used for a post-hoc analysis of association between hypercoagulability and thrombosis and death.

### 2.6. Ethics Statement

The study was conducted in accordance with the declaration of Helsinki. Written informed consent was obtained from all patients before any study-related procedures. The study protocol was approved on 30 June 2020 by the Independent Ethical Committee of Dmitry Rogachev National Medical Research Center of Pediatric Hematology, Oncology, and Immunology, Ministry of Healthcare of Russia, Moscow (approval №: 4/2020, issued by Rumyantsev AG and Ataullakhanov FI). The study was carried out as part of the COVID-19-Associated Coagulopathy Predicted by Thrombodynamic Markers Clinical Trial (CoViTro-I; ClinicalTrials.gov NCT-05330832). All data were impersonalized prior to statistical analysis. Data are available on https://github.com/shakhidzhanov-s/COVITRO/ (accessed on 10 March 2025).

### 2.7. Statistical Analysis

The statistical analysis of continuous data was conducted using the two-sided Mann–Whitney U test with continuity correction. We did not check the distributions for normality since this alters the conditional Type I error rates [44]. For categorical data, analysis was performed using the two-sided Fisher’s exact test. A *p*-value <0.05 was considered statistically significant. The level of significance denotations are as follows: *—*p*-value < 0.05, **—*p*-value < 0.01, ***—*p*-value < 0.001. No corrections for multiple hypothesis testing were applied. Outliers were not excluded before the analysis. Density distributions were obtained using the non-parametric kernel density estimation method. No imputation methods were used to address missing data.

The Kaplan–Meier method was used to analyze time-to-event data. Continuous variables were standardized before analysis. The endpoints were death and thrombosis (venous or arterial). For each endpoint, values of TDX-V, D-dimer, and APTT before the event were analyzed as independent variables. Patients who did not reach the endpoint due to being discharged were right-censored. Cox regression was performed to estimate risk ratios under the assumption that risks were proportional; we found that the risks remained relatively constant in patients hospitalized up to 45 days, and their data were analyzed. We also did not analyze the data of patients who were hospitalized for less than five days and of those who had thrombosis before the third day of hospitalization. The data of patients with missing values were not analyzed. Patients’ age, sex, and body mass index were chosen as confounders.

ROC analysis was conducted to evaluate the specificity and sensitivity of various coagulation assays, and to develop optimal models for predicting thrombosis and mortality using different combinations of coagulation markers.

The analysis was performed and graphs were plotted using the R programming language [45].

## 3. Results

Of the 4032 patients, 3860 met the inclusion criteria (Figure 1). The study cohort’s characteristics are given in Table 1.

The study cohort consisted of 51.1% women and 48.9% men, with a median patient age of 64 years and a median body mass index of 28.5. The majority of patients (94.6%) were admitted from home via emergency medical services (ambulance), while 5.4% were transferred from other hospitals. During the course of treatment, 42.8% of patients required admission to the intensive care unit (ICU), and 22.1% died. The median duration of hospitalization was 10 days (Table 1).

Upon admission, patients exhibited decreased SpO_2_ levels (median 94%), and 52.3% required oxygen support (Table 1). The majority of patients presented with CT severity scores of CT1 (36.0%) and CT2 (28.7%). The laboratory findings revealed elevated D-dimer levels (721 ng/mL), slightly prolonged prothrombin time (13.2 s), and increased fibrinogen levels (5.6 g/L). Additionally, levels of glucose (6.4 mmol/L), lactate dehydrogenase (324.7 AU/L), and C-reactive protein (60.4 mg/L) were elevated. The estimated glomerular filtration rate (CKD-EPI) was slightly below normal, at 56.8 mL × min^−1^ × 1.73 m^−2^.

Regarding treatments, 13.3% of patients received IL6/IL6R blockers, 13.5% received steroids, 3.2% received JAK inhibitors, and 11.3% received antibiotics (Table 1).

Many patients had cardiovascular comorbidities (Table 1). Venous thromboembolism (VTE) occurred in 474 patients, with deep vein thrombosis being the most frequent form (20.1%). Arterial thromboembolism (ATE) occurred in 79 patients, with limb artery thrombosis being the most frequent form (1.9%, Table 1).

We defined severe illness as the need for admission to the ICU; of our patients, 1654 were admitted to the ICU during their hospitalization, and were classified as severely ill. Patients who did not require ICU admission were classified as mildly ill. The characteristics of severely ill and mildly ill patients are given in Table 2.

Severely ill patients were older (68 vs 62), and had a higher rate of VTE and arterial thromboembolism (ATE), a more pronounced inflammatory state (CRP 98.8 vs 40.1 mg/L, white blood cell count 7.6 vs 5.8 × 10^9^/L, metabolic (glucose 7.0 vs 6.0 mmol/L) and coagulation (D-dimer 1256 vs 487 ng/mL) disturbances (Table 2). Severely ill patients also had higher rates of cardiovascular comorbidities, such as coronary artery disease, arterial hypertension, diabetes mellitus and chronic kidney disease. Severely ill patients also had higher computed tomography scores and required higher levels of respiratory support at admission. Among the severely ill, 25.3% received IL6/IL6R blockers, 2.4% received JAK inhibitors, 25.6% received steroids. Severely ill patients had higher rates of VTE and ATE (Table 2).

### 3.1. Heparins Treated Baseline Hypercoagulability in the Majority of Patients

Thrombodynamics clot growth rates (TDX-Vs) were measured at admission before LMWH administration. On the subsequent days, TDX-Vs were measured at the trough levels of LMWH pharmacological activity. The proportions of patients with hypercoagulability, normal coagulability, and hypocoagulability at these time points are given in Table 3.

At admission, hypercoagulability (TDX-V >29 µm/min) was present in 75.6% of severely ill and 79.6% of mildly ill patients. By the second day, the proportion of patients with hypercoagulability decreased to 28.0% in the group of severely ill patients and to 25.6% in the group of mildly ill patients. At the same time, the proportion of patients with hypocoagulability (TDX-V < 20 µm/min) increased from 7.1% to 53.4% in the severely ill group, and from 3.1% to 42.1% in the mildly ill group.

These changes in proportions were observed only up to the second day of hospitalization; in the subsequent days, the proportions of patients with hypercoagulability, hypocoagulability, and normal coagulability remained almost identical to those measured on day 2 (Table 3).

This shows that LMWH exerted its entire treating effect within the first two days of hospitalization—the hypercoagulability present at admission was resolved in the majority of patients. However, in about one-quarter of patients, it persisted despite anticoagulation.

### 3.2. Escalated Doses of Heparins Did Not Prevent Intense Hypercoagulability in Severely Ill Patients

We obtained TDX-V distributions for patients receiving prophylactic, intermediate, and therapeutic doses of heparins (Figure 3). In both severely ill and mildly ill patients receiving LMWHs, the main peaks and medians of the distributions shifted towards lower TDX-V values as the dose increased (Figure 3A,B). A similar trend was observed with UFH, where higher doses also resulted in a shift of the distributions towards lower TDX-V values (Figure 3C).

However, the distributions of severely ill and mildly ill patients receiving LMWH featured a notable difference (Figure 3A,B)—the distributions of severely ill patients had tails in the range of extremely high clot growth rates (TDX-V ≥ 50 µm/min). In contrast, the distributions of mildly ill patients mostly lacked such tails. Further, we refer to the range TDX-V ≥ 50 µm/min as intense hypercoagulability.

As can be seen from Figure 3B, these intense hypercoagulability tails did not shift towards lower TDX-V values, and their size did not decrease when escalated LMWHs were used, suggesting no reduction in the risk of intense hypercoagulability at higher doses. This contrasts with the main peaks of the distributions, which shifted towards lower TDX-V as higher LMWH doses were used. This might suggest that LMHW had a dose-dependent effect on hypercoagulability only when the TDX-V values were below 50 µm/min. In other words, LMWHs reduced the risks of hypercoagulability only if it was not intense.

On the other hand, since mildly ill patients did not have intense hypercoagulability tails, their risk of hypercoagulability decreased when the LMWH doses were increased; the escalation shifted the TDX-V distribution towards lower values.

The distributions of severely ill patients receiving UFH also had intense hypercoagulability tails (Figure 3C). However, patients receiving UFH via infusion (therapeutic dose) had intense hypercoagulability less often than those receiving prophylactic or intermediate UFH subcutaneously, as can be seen from the sizes of the tails.

In our study, 28.3% of severely ill patients had intense hypercoagulability while receiving therapeutic doses of heparins. We investigated whether this arose from some patients with particular clinical profiles. We examined the influence of demographic characteristics on the TDX-V distributions of severely ill patients receiving therapeutic doses of heparins (Appendix A); the distributions remained almost unchanged, with intense hypercoagulability tails consistently present irrespective of hypertension, diabetes, CCI+CVD, heart failure, coronary artery disease, peripheral atherosclerosis, atrial fibrillation, cancer, sex, or CT score at admission. They also did not depend on age and BMI, although intense hypercoagulability was slightly less pronounced in patients aged 18–44. The distributions were also unaffected by the exclusion of patients transferred from other hospitals from the sample (Appendix A).

### 3.3. Intense Hypercoagulability Was Temporary

Next, we analyzed individual TDX-V time courses, and typical examples are shown in Figure 4A. The figure shows data from six patients—five severely ill (#1–5) and one mildly ill patient (#6). The blood coagulation state of severely ill patients often evolved during treatment, sometimes abruptly (exemplified by patients #1, 2, and 4). Patients #1–4 also illustrate that intense hypercoagulability was often transient; their TDX-V values were ≥50 µm/min at the certain points of time (e.g., from day 3 to day 10 and on day 12 in patient #1). During some of these periods, thrombotic complications developed in these patients. However, on the other days, these same patients exhibited TDX-V values < 50 µm/min, including normal or even hypocoagulable TDX-V values.

In contrast, severely ill *patient #5* had a more stable blood coagulation state and did not have intense hypercoagulability. This patient had prolonged hypocoagulability, likely contributing to the development of gastrointestinal bleeding. Mildly ill patients usually showed little to no variation in their coagulation state throughout hospitalization, as exemplified by *patient #6*.

Figure 4B shows patients’ TDX-V averages for two groups of severely ill individuals: those who had periods of intense hypercoagulability (grey box) and those who did not (red box). Severely ill patients with periods of intense hypercoagulability had higher TDX-V averages compared to those without such periods. However, the TDX-V averages of severely ill patients with periods of intense hypercoagulability were below 50 μm/min (the grey box is below 50 μm/min). This indicates that, on average, these patients spent a significant portion of their hospitalization time exhibiting coagulation states other than intense hypercoagulability. In other words, intense hypercoagulability was a temporary state for the majority of severely ill patients.

We did not find a correlation between intense hypercoagulability in severely ill patients and low antithrombin III plasma levels—additional examples of TDX-V and antithrombin III levels’ time courses are shown in Appendix A.

### 3.4. Severely Ill Patients with Intense Hypercoagulability Had Higher Levels of D-Dimer, Inflammation Markers and Better Glomerular Filtration Rates

We examined which laboratory parameters differed between severely ill patients who had intense hypercoagulability and those who did not have it. The parameters having significant differences are shown in Figure 5.

Severely ill patients with intense hypercoagulability had higher D-dimer levels, indicating increased thrombosis rates, as well as elevated levels of lactate dehydrogenase activity in plasma (LDH), C-reactive protein (CRP), and white blood cell counts (WBC). They also required higher levels of respiratory support. Interestingly, these patients had higher CKD-EPI, which may reflect the more efficient clearance of heparins than in patients without hypercoagulability. Similar results were obtained using the Cockroft and Gault formula; according to it, the medians were 91.2 and 83.6 mL × min-1 × 1.73 m^−2^ in patients with and without intense hypercoagulability, respectively (*p* < 0.01).

### 3.5. Intense Hypercoagulability Was a Risk Factor for Thrombosis and Death

We studied how intense hypercoagulability affected the risks of thrombosis and death in severely ill patients. A Kaplan–Meier analysis showed that the more persistent intense hypercoagulability was, the higher the risks of thrombotic complications and mortality (Figure 6). A Cox regression showed that severely ill patients with intense hypercoagulability faced a 1.68-fold higher hazard rate for death and a 3.19-fold increased hazard rate for thrombosis if intense hypercoagulability persisted for more than 25% of the time prior to the event (Table 4); these figures doubled if it persisted for more than 50% of the time prior to the event.

We performed similar analyses for D-dimer and APTT. Severely ill patients with elevated D-dimer levels (≥5000 ng/mL) had a 1.43-fold higher hazard rate for death and a 1.90-fold higher hazard rate for thrombosis if intense hypercoagulability persisted for more than 25% of the time prior to the event (Appendix A); these figures doubled if it persisted for more than 50% of the time prior to the event. On the other hand, shortened APTT (≤25.1 s) was not associated with increased risks of thrombosis or mortality (Appendix A).

We also found that new thrombotic complications were often preceded by high levels of TDX-V and D-dimer (Appendix A). Meanwhile, APTT was not shortened at these moments.

In our cohort, hypocoagulability preceded new thrombotic complications only in seven patients (8.2%). In three cases, platelet counts exceeded the normal range; in two cases, there was a sharp drop in platelet levels prior to thrombosis. In the remaining two cases, platelet counts were below normal.

### 3.6. Combining TDX-V and D-Dimer Assays Can Enhance the Accuracy of Predicting Thrombotic Events and Fatal Outcomes

Peak levels of D-dimer and TDX-V could be used to assess the risk of thrombosis: the ROC-AUC for D-dimer and TDX-V levels were 0.710 and 0.709, respectively (Appendix A). However, an index combining both TDX-V and D-dimer had higher ROC-AUC = 0.765. Nadir APTT values had no prognostic value (ROC-AUC = 0.534). Similarly, an index combining both TDX-V values and peak D-dimer levels demonstrated the best prognostic accuracy for predicting lethal outcomes, with ROC-AUC = 0.831 (Appendix A). These findings suggest that taking into account both D-dimer and Thrombodynamics data may provide more accurate assessments of the risks of thrombotic complications and fatal outcomes.

## 4. Discussion

We examined blood coagulability in 2206 mildly ill and 1654 severely ill COVID-19 patients using the APTT, D-dimer, and Thrombodynamics assays. Patients in both groups received prophylactic, intermediate and therapeutic doses of heparins.

The characteristics of our study cohort were consistent with those reported in other studies; our severely ill patients had comorbidities (e.g., diabetes, hypertension, coronary artery disease, chronic kidney disease), which is typical [46]. Also, at admission, patients had elevated levels of D-dimer, fibrinogen, glucose, lactate dehydrogenase, and C-reactive protein, and these markers were notably higher in severely ill patients, which is also typical [47,48]. Prothrombin time was slightly elevated above the normal range, while APTT remained within the reference range, which is consistent with another study [49]. The overall incidences of venous and arterial thromboembolic complications in our study are comparable to those in previously published data [50,51,52,53].

We assessed how effectively the Thrombodynamics clot growth rate could be used to evaluate hypercoagulability in COVID-19 patients. We showed that severely ill patients having TDX-V ≥ 50 µm/min had a 1.68-fold higher hazard rate for death and a 3.19-fold higher hazard rate for thrombosis than severely ill patients without such high TDX-V values. We also found that peak TDX-V values had good prognostic accuracy in assessing the risks of thrombotic complications and lethal outcomes. This suggests that the Thrombodynamics assay accurately detects hypercoagulability in COVID-19 patients.

We evaluated the effectiveness of using D-dimer to assess hemostasis in order to compare its performance with that of TDX-V. We found that elevated D-dimer levels also indicate high risks of thrombotic complications and fatal outcomes, which aligns well with the published data [54]. We also showed that the concurrent consideration of Thrombodynamics and D-dimer results enhances the accuracy when assessing the risks of thrombosis and fatal outcomes.

The D-dimer test has several limitations that must be considered when assessing coagulation. Firstly, elevated D-dimer levels do not necessarily indicate an ongoing risk of thrombotic complications. D-dimer levels rise following thrombus formation and the initiation of fibrinolysis [55]. However, by this point, the patient may have already transitioned out of a hypercoagulable state, and thus may no longer be at significant risk for further thrombosis. Secondly, the test cannot guide adjustments to anticoagulation therapy because it lacks sensitivity to hypocoagulable states. Thirdly, its reliability may be compromised in severely ill COVID-19 patients experiencing fibrinolysis shutdown [56]. Therefore, it is crucial to supplement D-dimer testing with other coagulation assays that do not share these limitations.

APTT is used to monitor heparin therapy and detect deficiencies in coagulation factors. While shortening of the APTT is rare and not typically considered a marker of hypercoagulability, it can be used for this purpose in some cases [57]. APTT in COVID-19 patients is generally within the normal range, or slightly prolonged [58]. We found that APTT shortening in severely ill patients did not indicate a higher risk of thrombosis; hence, it did not detect hypercoagulability. The low performance of APTT in the detection of hypercoagulability is most likely associated with the high concentrations of activators used in it [59,60]. As a result, it is sensitive to factor deficiencies, but is poorly suited for detecting hypercoagulability.

Some studies [61,62] have suggested that, in COVID-19 patients, an elevated risk of mortality may be indicated by prolonged APTT rather than shortened APTT. Prolonged APTT may result from a deficiency in clotting factors due to consumptive coagulopathy, which is preceded by a hypercoagulable state. However, since this test is not well-suited for detecting hypercoagulability [59,60], APTT only becomes sensitive at advanced stages of the disease, when treatment may no longer be effective. Another potential cause of prolonged APTT could be antiphospholipid syndrome, which has been observed in some patients with COVID-19 [63].

A potential concurrent factor causing intense hypercoagulability could be the high level of factor VIII, which was observed in some patients with coronavirus infection [61,64]. The study [54] showed that patients with factor VIII levels >314 IU/dl had 16.6-fold higher hazard rates for death. Intense hypercoagulability can also be caused by the massive cell-release of procoagulant extracellular vesicles induced by severe inflammation [6,7].

In our study, antithrombin III levels were generally within the normal range, and were not associated with episodes of intense hypercoagulability. Other studies also report almost normal antithrombin III levels [26,27]. We suggest that intense hypercoagulability did not result from low antithrombin III levels. Also, intense hypercoagulability was not correlated with transfusions of convalescent or fresh-frozen plasma in our study.

In addition to the blood coagulation, platelets may play a significant role in the development of thrombotic complications during coronavirus infection. Their hyperreactivity could drive persistent hypercoagulability in patients in our study. For instance, large platelet aggregates were detected in high numbers in the blood of patients [65]. In another study, using machine learning methods and single-cell transcriptomics, distinct platelet subpopulations that correlate with disease severity were identified [31]. In another study, a disruption in the gene expression profile of platelets was observed, along with increased platelet hyperactivity [32]. On the other hand, some research [36,37] suggests that platelet activity is reduced in COVID-19 patients, with this suppression being more pronounced in critically ill patients [36]. It is also worth noting that clinical trials of antiplatelet drugs did not demonstrate their effectiveness in reducing the risks of fatal outcomes and thrombosis [17,38,39]. However, it is also possible that platelets play an important role in the propagation of coronavirus infection [33]. In our study, we did not examine the role of platelets, but we acknowledge that they could have played an important role in the development of thrombotic complications in some of our patients.

The risk of a fatal outcome may increase in cases of coronavirus infection not only due to abnormalities in the plasma or platelet components of hemostasis, but also as a result of impaired immune system function [66]. Currently, there are numerous models for assessing the risk of severe illness and death in such patients, including those with severe illness [67,68,69,70]. In addition to standard coagulation test values, these models often include indicators such as the level of C-reactive protein, the neutrophil-to-lymphocyte ratio, and erythrocyte sedimentation rate [71].

In prior studies, Thrombodynamics and Thromboelastography assays showed that most patients were hospitalized with hypercoagulability, but it was not present at the peak pharmacological activity of LMWH [72]. In our study, we found that following the initiation of treatment and anticoagulant therapy, approximately 70% of patients experienced the normalization of blood coagulation, or in some cases, even developed hypocoagulability at the trough levels of LMWH pharmacological activity. These results align with those obtained from the Thrombin Generation assay (TG), which showed that COVID-19 patients have hypercoagulability at admission, but that standard thromboprophylaxis reduced TG to levels of healthy controls at the trough level of anticoagulation [73,74].

The randomized clinical trials have shown that dose escalation does not reduce mortality in severely ill patients, and its effect on the thrombosis rate is unclear [17,18,19,20,21,22,23,24,25]. We hypothesize that severely ill patients may experience hypercoagulability even when receiving escalated doses of heparins. We studied the effects of various doses of heparins on blood coagulation in severely ill and mildly ill patients using the Thrombodynamics assay.

Three key observations were made. Firstly, heparins had a dose-dependent effect on blood coagulation in both severely ill and mildly ill patients when TDX-V was below 50 μm/min or UFH infusion was used. A similar dose-dependent effect was also observed in the TG assay [74].

Secondly, we found that some severely ill patients may exhibit extremely high clot growth rates (TDX-V ≥ 50 μm/min) at the trough level of pharmacological activity, even if doses of LMWH were escalated. Moreover, LMWH did not exert a dose-dependent effect on intense hypercoagulability in severely ill patients. Hence, the risks of thrombosis and death were not reduced in severely ill patients if they had intense hypercoagulability and received escalated LMWH.

In contrast, mildly ill patients did not have intense hypercoagulability. Dose escalation shifted their TDX-V distribution towards lower values, and therefore, their risk of hypercoagulability decreased when LMWH doses were escalated. This may explain why escalated doses were effective in mildly ill patients and ineffective in severely ill patients [17,18,19,20,21,22,23,24,25,29,30].

Thirdly, plasma coagulability in severely ill patients was highly dynamic, and the same patient could experience periods of intense hypercoagulability that evolved to other coagulation states.

Severely ill COVID-19 patients can experience a strong inflammatory process leading to blood coagulation disturbance [75]. This can explain the episodes of intense hypercoagulability in severely ill patients with higher levels of CRP, WBC, and LHD. Another observation was that patients with episodes of intense hypercoagulability had better CKD-EPI. This might suggest lower trough pharmacological activity levels of heparins in them, which could allow the development of intense hypercoagulability, even if the patient received escalated doses of anticoagulation [76]. It is possible that the more frequent administration of LMWH without daily-dose escalation could be more effective in the prevention of intense hypercoagulability; however, this hypothesis needs further validation.

Alternative anticoagulation strategies, such as transitioning to UFH infusion, might be considered for patients with TDX-V >50 μm/min. Our study has demonstrated that UFH infusion was associated with fewer instances of intense hypercoagulability. Therefore, TDX-V values exceeding 50 μm/min could serve as a clinical decision point for switching to UFH infusion, with the rate adjusted using APTT or Thrombodynamics [43] (Table 5).

### 4.1. Study Limitations

One of the key limitations of our study is the absence of anti-factor Xa activity measurements, which are commonly used to assess the efficacy of LMWH therapy. The lack of Anti-Xa data leaves a gap in validating LMWHs’ effectiveness in our research. Combining these measurements with Thrombodynamics could provide a more precise evaluation of hypercoagulability, helping to determine whether it results from the insufficient pharmacological activity of the anticoagulant. On the other hand, Anti-factor Xa activity alone is insufficient for a comprehensive assessment of thrombotic complication risk [77,78], underscoring the need to complement it with additional blood coagulation assays, such as Thrombodynamics. It is worth noting that Thrombodynamics demonstrates sensitivity to heparin comparable to that of anti-factor Xa activity [43].

In our study, 8.2% of venous thrombotic events occurred when TDX-V showed hypocoagulability. Currently, we cannot explain this. Possibly, these events could result from platelet hyperreactivity or thrombocytosis. Hence, platelet activity assays might be necessary to get a full picture of clotting state in COVID-19 patients. The lack of these data also limits out study.

During the study, we frequently saw rapid fluctuations in the blood coagulation state of severely ill patients. Currently, we do not have an explanation of what can induce these rapid changes.

We chose to analyze venous and arterial thromboses collectively, as TDX-V values were elevated prior to both types of events. Our results provide an average risk estimate for thrombotic events, encompassing both venous and arterial occurrences. However, it is important to note that the risks for venous thrombosis may differ from those for arterial thrombosis.

We were unable to obtain baseline TDX-V values in patients who received their first anticoagulation treatment in an ambulance. This practice was introduced midway through the study; however, by that time, we had already collected a substantial number of baseline TDX-V values, which we believe adequately reflected the baseline blood coagulability of the entire cohort. Additionally, a few patients had significant gaps between TDX-V measurements, and were therefore excluded from TDX-V-related analyses. Nevertheless, these gaps were not associated with illness severity, and we do not believe they impacted the overall validity of our results.

The study extended over a year and included the beta, gamma, and delta variants of COVID-19, each of which could affect hemostasis differently. We were unable to determine the virus variants in our patients for comparison with our data and for the validation of our results for different virus variants. Also, whether these results are valid in relation to the subsequent waves and the current continuous level of COVID-19 episodes is not known.

### 4.2. Conclusions

The pharmacological effect of LMWHs at the trough level might be too low to prevent thrombosis in severely ill patients with severe inflammation and better creatinine clearance, even if escalated doses are used. Thrombodynamics assays can reliably detect hypercoagulability in COVID-19 patients. The simultaneous consideration of Thrombodynamics and D-dimer results further enhances the accuracy of assessing the risks of thrombosis and fatal outcomes.

## Figures and Tables

**Figure 1 jcm-14-01966-f001:**
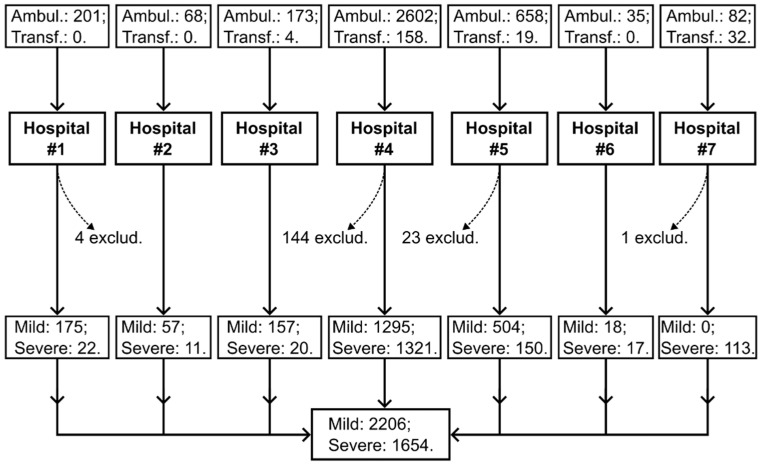
Distribution of hospitalized patients by admission route, hospitals, and severity of illness. The study was conducted in seven hospitals. The majority of patients were admitted from home via emergency medical services (ambulance, *n* = 3819), while another group of patients was transferred from other hospitals (*n* = 213). For 172 patients, COVID-19 was not confirmed, and they were excluded from the study. Ambul.—ambulance, Transf.—transferred.

**Figure 2 jcm-14-01966-f002:**
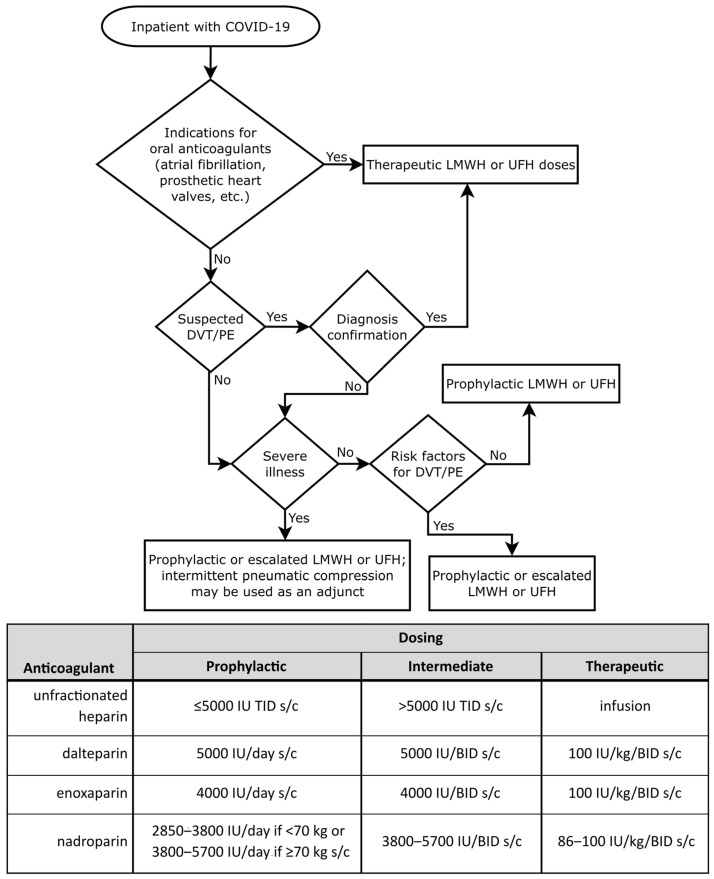
Algorithm for the use of LMWHs and UFH in the treatment of COVID-19 in adult patients in an inpatient setting. Anticoagulation de-escalation was recommended for patients with kidney injury. It could also be temporarily lowered or discontinued in cases of bleeding, intubation, or surgical intervention. A table below the flowchart shows the doses of heparins used during the study. LMWH—low-molecular-weight heparin, UFH—unfractionated heparin, DVT—deep vein thrombosis, PE—pulmonary embolism, s/c—subcutaneously, TID—three time a day, BID—two times a day.

**Figure 3 jcm-14-01966-f003:**
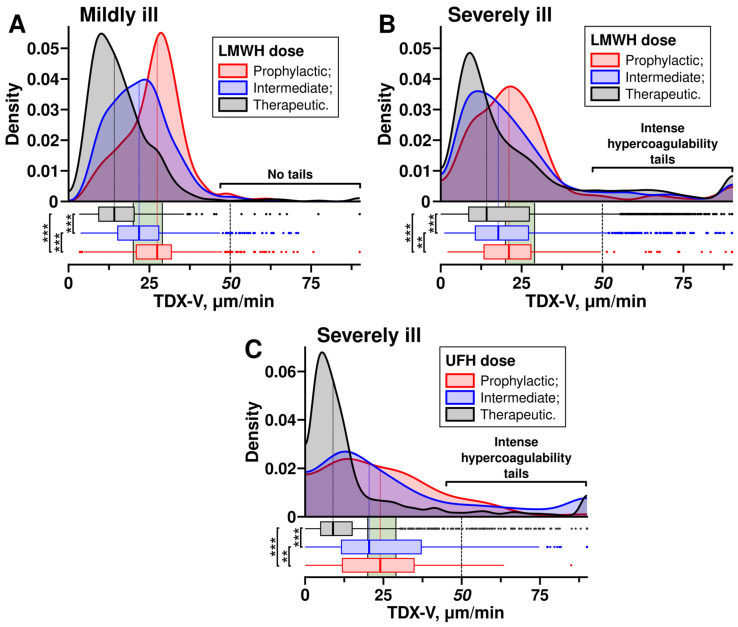
TDX-V distributions of mildly ill and severely ill patients receiving different doses of heparins. LMWH—low-molecular-weight heparin, UFH—unfractionated heparin. Thrombodynamics clot growth rates (TDX-V). The distributions show TDX-V values obtained from the second day of hospitalization (since TDX-V values at admission were obtained before a patient was given LMWH or UFH). Patients received prophylactic, intermediate, or therapeutic doses of heparins. Samples were collected at the trough level of heparins’ pharmacological activity (excluding UFH infusion, which was collected at least 6 h after the bolus or the dose adjustment). Boxes under the distributions show their medians and quartiles. Green areas show the normal range (20–29 µm/min). Statistical significance was assessed with the Mann–Whitney U test; levels of significance are shown to the left of the boxes. **—*p*-value < 0.01, ***—*p*-value < 0.001. (**A**,**B**) Distributions in mildly ill and severely ill patients receiving different doses of LMWH. (**A**) Prophylactic—458 patients with a total of 1035 measurements, intermediate—558 patients with a total of 1526 measurements, therapeutic—259 patients with a total of 593 measurements. (**B**) Prophylactic—144 patients with a total of 327 measurements, intermediate—452 patients with a total of 1281 measurements, therapeutic—882 patients with a total of 3939 measurements. (**C**) Distributions in severely ill patients receiving different doses of UFH. Distributions of mildly ill patients receiving UFH are not shown due to a small number of cases. Prophylactic—42 patients with a total of 79 measurements, intermediate—122 patients with a total of 352 measurements, therapeutic—245 patients with a total of 1151 measurements.

**Figure 4 jcm-14-01966-f004:**
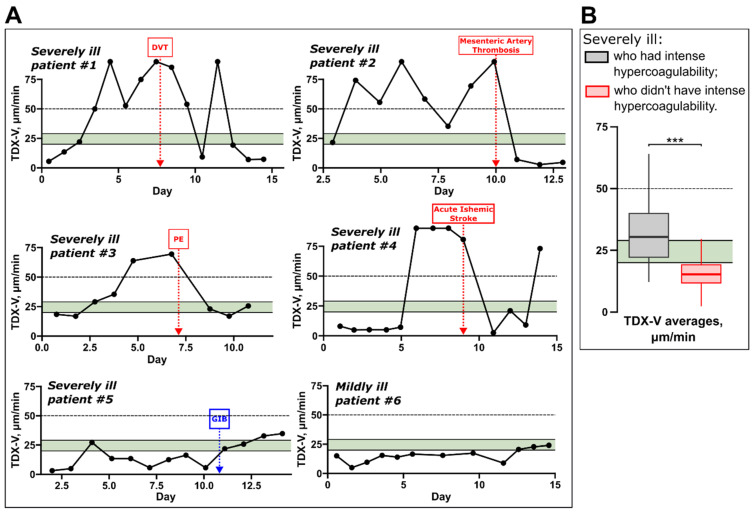
TDX-V of severely ill patients who experienced intense hypercoagulability. (**A**) Examples of TDX-V time courses. Five severely ill (#1–5) and one mildly ill patient (#6) are shown. Red boxes show the moments at which thrombotic complications were confirmed by instrumental methods. DVT—deep vein thrombosis, PE—pulmonary embolism. Dashed horizontal lines show the lower limit of intense hypercoagulability. Patient #5 had gastrointestinal bleeding on day 11. (**B**) TDX-V averages of severely ill patients who had periods of intense hypercoagulability (grey) and those who did not have them (red). Statistical significance was assessed with the Mann–Whitney U test, *n* = 312 (grey box) and 492 (red box). ***—*p*-value < 0.001.

**Figure 5 jcm-14-01966-f005:**
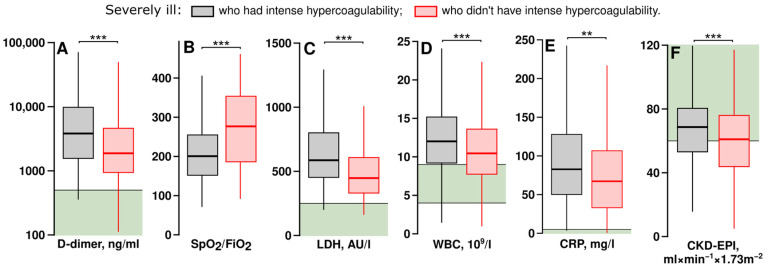
Laboratory averages of severely ill patients who had intense hypercoagulability and those without it. The values of severely ill patients receiving therapeutic doses of heparins were analyzed. Black boxes show the averages of severely ill patients who had intense hypercoagulability, red boxes show the averages of severely ill patients without intense hypercoagulability. Green areas show the normal ranges. Statistical significance was assessed with the Mann–Whitney U test. **—*p*-value < 0.01, ***—*p*-value < 0.001. (**A**) D-dimer level, the logarithmic scale is used, *n* = 219 (grey box) and 308 (red box). (**B**) SpO_2_ to FiO_2_ ratio, *n* = 308 (grey box) and 580 (red box). (**C**) Lactate dehydrogenase activity, *n* = 212 (grey box) and 340 (red box). (**D**) White blood cell count, *n* = 339 (grey box) and 629 (red box). (**E**) C-reactive protein level, *n* = 339 (grey box) and 625 (red box). (**F**) Chronic kidney disease–epidemiology collaborative group equation, *n* = 298 (grey box) and 620 (red box).

**Figure 6 jcm-14-01966-f006:**
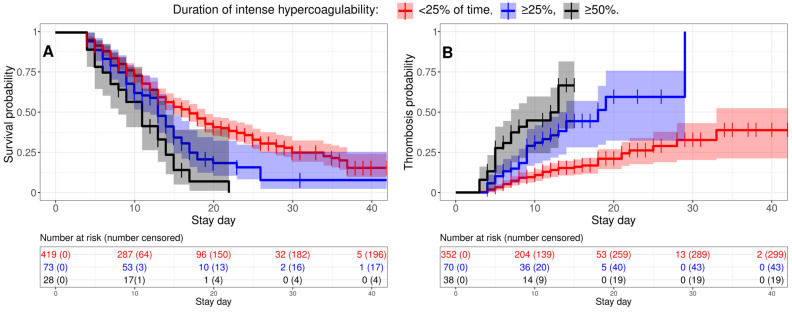
Intense hypercoagulability increases the risks of death and thrombosis in severely ill patients. Kaplan–Meier curves are shown for survival (**A**) and thrombosis (**B**) probabilities in severely ill patients with different durations of intense hypercoagulability (<25% of the time before the event (death in (**A**) and thrombosis in (**B**))—red line, ≥25%—blue, and ≥50%—black).

**Table 1 jcm-14-01966-t001:** Patients’ characteristics.

Variable	Total (*n* = 3860)
Age, median (Q_1_–Q_3_)	64 (54–74)
Female, % (*n*)	51.1 (1974)
Referred from another hospital, % (*n*)	5.4 (210)
Body mass index, median (Q_1_–Q_3_)	28.5 (25.2–33.1)
Death, % (*n*)	22.1 (853)
Admitted to ICU, % (*n*)	42.8 (1654)
Length of stay in days, median (Q_1_–Q_3_)	10 (7–15)
Vitals on admission, median (Q_1_–Q_3_)	
SPO_2_, %	94 (92–96)
Respiratory rate, 1/min	22 (20–24)
Systolic pressure, Hg	130 (117–140)
Diastolic pressure, Hg	80 (70–87)
Heart rate, 1/min	91 (81–101)
Respiratory support on admission, % (*n*)	
ECMO	0.0 (1)
Invasive ventilation	3.5 (122)
Non-invasive ventilation	7.4 (256)
Nasal oxygen	41.4 (1432)
No oxygen therapy	47.7 (1649)
Computer tomography score on admission, % (*n*)	
CT0	8.1 (311)
CT1	36.0 (1388)
CT2	28.7 (1107)
CT3	20.1 (776)
CT4	7.1 (275)
Laboratory on admission, median (Q_1_–Q_3_)	
TDX-V, 20–29 µm/min	34.8 (29.7–43.8) *n* = 1370
TDX-TSP, >30 min	21.9 (17.0–26.0) *n* = 545
D-dimer, <500 ng/mL	721 (355–1529) *n* = 2753
APTT, 25.1–36.5 s	30.5 (27.9–33.5) *n* = 3220
PT, 9.4–12.5 s	13.2 (12.2–14.6) *n* = 3288
Fibrinogen, 2–4 g/L	5.6 (4.4–6.9) *n* = 3074
Hemoglobin, 120–160 g/L	136 (124–148) *n* = 3624
Platelet count, 180–320 × 10^9^/L	202 (156–259) *n* = 3625
White blood cell count, 4–9 × 10^9^/L	6.4 (4.8–9.2) *n* = 3624
Creatinine, 49–104 µmol/L	95.3 (80.0–114.0) *n* = 3610
Glucose, 4.1–5.9 mmol/L	6.4 (5.5–7.8) *n* = 3576
Alanine aminotransferase, <50 AU/L	29.4 (19.0–48.0) *n* = 3581
Aspartate aminotransferase, <50 AU/L	37.0 (26.3–56.0) *n* = 3583
Lactate dehydrogenase, <250 AU/L	324.7 (246.4–486.4) *n* = 1839
Bilirubin, 5–21 µmol/L	11.1 (8.2–15.0) *n* = 3548
C-reactive protein, <5 mg/L	60.4 (21.0–118.9) *n* = 3609
CKD-EPI, >60 mL × min^−1^ × 1.73 m^−2^	56.8 (43.5–69.4) *n* = 3606
Medication during hospitalization, % (*n*)	
Low-molecular-weight heparins	94.2 (3635)
Unfractionated heparin	17.9 (691)
IL6/IL6R blockers	13.3 (512)
IL17 blockers	0.1 (4)
JAK inhibitors	3.2 (125)
Steroids	13.5 (520)
Antibiotics	11.3 (437)
Statins	10.2 (392)
Diuretics	15.3 (592)
Antiplatelet	3.8 (147)
Comorbidities, % (*n*)	
Peripheral atherosclerosis	4.8 (187)
Coronary artery disease	18.1 (700)
Heart failure	21.9 (844)
CCI+CVD	22.2 (856)
Atrial fibrillation	14.4 (555)
Hypertension	62.4 (2410)
Chronic kidney disease	14.9 (575)
Diabetes mellitus	23.4 (905)
Cancer	8.6 (333)
Acute kidney injury	10.9 (421)
Complications, % (*n*)	
Vein thromboembolism	
Deep vein thrombosis	10.0 (385)
Pulmonary embolism	2.5 (98)
Superficial vein thrombosis	2.6 (99)
Other vein thrombosis ^#^	0.2 (6)
Arterial thromboembolism	
Acute ischemic stroke	0.5 (20)
Mesenteric artery thrombosis	0.4 (14)
Limb artery thrombosis	0.9 (35)
Other artery thrombosis ^##^	0.2 (7)

^#^ Hepatic vein thrombosis, pampiniform plexus thrombosis, renal vein thrombosis, splenic vein thrombosis. ^##^ Internal carotid artery thrombosis, renal artery thrombosis, splenic artery thrombosis. CT—computed tomography, CKD-EPI—chronic kidney disease–epidemiology collaborative group equation, CCI+CVD—chronic cerebral ischemia + cerebrovascular disease, ECMO—extracorporeal membrane oxygenation.

**Table 2 jcm-14-01966-t002:** Comparison of severely ill and mildly ill patients.

Variable	Severely Ill (*n* = 1654)	Mildly Ill (*n* = 2206)	*p*-Value, Severely vs Mildly Ill
Death, % (*n*)	51.6 (853)	0 (0)	<0.0001 †
Age, median (Q_1_–Q_3_)	68 (58–79)	62 (51–71)	<0.0001
Female, % (*n*)	50.5 (835)	51.6 (1139)	0.49
Referred from another hospital, % (*n*)	9.9 (164)	2.1 (46)	<0.0001
Body mass index, median (Q_1_–Q_3_)	28.8 (25.3–33.7)	28.3 (25.2–32.3)	0.06
Length of stay in days, median (Q_1_–Q_3_)	13 (8–20)	9 (7–12)	<0.0001
Vitals on admission, median (Q_1_–Q_3_)			
SPO_2_, %	93 (90–95)	94 (93–96)	<0.0001
Respiratory rate, 1/min	24 (21–26)	22 (20–24)	<0.0001
Systolic pressure, Hg	129 (115–140)	130 (119–140)	0.95
Diastolic pressure, Hg	78 (70–85)	80 (70–88)	<0.0001
Heart rate, 1/min	92 (82–103)	91 (81–100)	0.016
Respiratory support on admission, % (*n*)			
ECMO	0.1 (1)	0.0 (0)	n.a.
Invasive ventilation	8.2 (122)	0.0 (0)	<0.0001 †
Non-invasive ventilation	17.0 (252)	0.2 (4)	<0.0001
Nasal oxygen	58.6 (871)	28.4 (561)	<0.0001
No oxygen therapy	16.2 (240)	71.4 (1409)	<0.0001
Computer tomography score on admission, % (*n*)			
CT0	7.4 (123)	8.5 (188)	0.23
CT1	23.7 (392)	45.2 (996)	<0.0001
CT2	26.0 (429)	30.8 (678)	0.001
CT3	27.9 (462)	14.2 (314)	<0.0001
CT4	14.9 (247)	1.3 (28)	<0.0001
Laboratory on admission, median (Q_1_–Q_3_)			
TDX-V, 20–29 µm/min	35.3 (29.1–44.6) *n* = 241	34.7 (29.8–43.7) *n* = 1129	0.84
TDX-TSP, >30 min	22.1 (16.4–26.3) *n* = 103	21.9 (17.4–25.9) *n* = 442	0.80
D-dimer, <500 ng/mL	1256 (686–2954) *n* = 1145	487 (277–906) *n* = 1608	<0.0001
APTT, 25.1–36.5 s	30.2 (27.0–33.6) *n* = 1497	30.7 (28.5–33.4) *n* = 1723	0.0002
PT, 9.4–12.5 s	13.5 (12.4–14.9) *n* = 1508	13.0 (12.0–14.3) *n* = 1780	<0.0001
Fibrinogen, 2–4 g/L	6.0 (4.5–7.4) *n* = 1441	5.3 (4.3–6.6) *n* = 1633	<0.0001
Haemoglobin, 120–160 g/L	134 (119–147) *n* = 1571	138 (127–149) *n* = 2053	<0.0001
Platelet count, 180–320 × 10^9^/L	191 (146–253) *n* = 1571	207 (165–264) *n* = 2054	<0.0001
White blood cell count, 4–9 × 10^9^/L	7.6 (5.4–11.4) *n* = 1571	5.8 (4.5–7.8) *n* = 2053	<0.0001
Creatinine, 49–104 µmol/L	99.0 (81.7–123.8) *n* = 1586	93.0 (79.3–110.0) *n* = 2024	<0.0001
Glucose, 4.1–5.9 mmol/L	7.0 (6.0–9.1) *n* = 1563	6.0 (5.3–7.0) *n* = 2013	<0.0001
Alanine aminotransferase, <50 AU/L	31.0 (20.0–52.0) *n* = 1573	28.0 (18.3–45) *n* = 2008	<0.0001
Aspartate aminotransferase, <50 AU/L	45.7 (31.0–68.8) *n* = 1575	32.0 (25.0–47.0) *n* = 2008	<0.0001
Lactate dehydrogenase, <250 AU/L	431.2, (298.9–610.0) *n* = 946	266.7, (219.8–344.0) *n* = 893	<0.0001
Bilirubin, 5–21 µmol/L	11.5 (8.4–16.3) *n* = 1568	10.8 (8.1–14.1) *n* = 1980	<0.0001
C-reactive protein, <5 mg/L	98.8 (41.1–170.1) *n* = 1579	40.1 (14.7–80.6) *n* = 2030	<0.0001
CKD-EPI, >60 mL × min^−1^ × 1.73 m^−2^	53.4 (38.7–67.1) *n* = 1584	59.3 (47.3–71.1) *n* = 2022	<0.0001
Medication during hospitalization, % (*n*)			
Low-molecular-weight heparins	93.1 (1540)	95.0 (2095)	0.72
Unfractionated heparin	39.4 (652)	1.8 (39)	<0.0001
IL6/IL6R blockers	25.3 (418)	4.3 (94)	<0.0001
IL17 blockers	0.0 (0)	0.2 (4)	n.a.
JAK inhibitors	2.4 (39)	3.9 (86)	0.008
Steroids	25.6 (424)	4.4 (96)	<0.0001
Antibiotics	14.8 (244)	8.7 (193)	<0.0001
Statins	14.7 (243)	6.8 (149)	<0.0001
Diuretics	20.6 (341)	11.4 (251)	<0.0001
Antiplatelet	5.6 (93)	2.4 (54)	<0.0001
Comorbidities, % (*n*)			
Peripheral atherosclerosis	7.7 (128)	2.7 (59)	<0.0001
Coronary artery disease	27.9 (461)	10.8 (239)	<0.0001
Heart failure	33.1 (548)	13.4 (296)	<0.0001
CCI+CVD	37.2 (615)	10.9 (241)	<0.0001
Atrial fibrillation	20.8 (344)	9.6 (211)	<0.0001
Hypertension	78.1 (1291)	50.7 (1119)	<0.0001
Chronic kidney disease	23.6 (390)	8.4 (185)	<0.0001
Diabetes mellitus	30.8 (509)	18.0 (396)	<0.0001
Cancer	12.0 (198)	6.1 (135)	<0.0001
Acute kidney injury	25.3 (419)	0.2 (5)	<0.0001
Complications, % (*n*)			
Vein thromboembolism			
Deep vein thrombosis	20.1 (332)	2.4 (53)	<0.0001
Pulmonary embolism	5.7 (94)	0.2 (4)	<0.0001
Superficial vein thrombosis	5.1 (84)	0.7 (15)	<0.0001
Other vein thrombosis ^#^	0.3 (5)	0.0 (1)	0.09
Arterial thromboembolism			
Acute ischemic stroke	1.2 (20)	0.0 (0)	<0.0001 †
Mesenteric artery thrombosis	0.8 (14)	0.0 (0)	<0.0001 †
Limb artery thrombosis	1.9 (32)	0.1 (3)	<0.0001
Other artery thrombosis ^##^	0.4 (7)	0.0 (0)	n.a.

†—If any group had zero events, we adjusted the calculation by substituting 1 for the 0 in that group to assess significance. ^#^ Hepatic vein thrombosis, pampiniform plexus thrombosis, renal vein thrombosis, splenic vein thrombosis. ^##^ Internal carotid artery thrombosis, renal artery thrombosis, splenic artery thrombosis. CT—computed tomography, CKD-EPI—chronic kidney disease–epidemiology collaborative group equation, CCI+CVD—chronic cerebral ischemia + cerebrovascular disease, ECMO—extracorporeal membrane oxygenation, n.a.—not assessed.

**Table 3 jcm-14-01966-t003:** Proportions of patients with hypercoagulability (TDX-V >29 µm/min), normal coagulability (20–29 µm/min), and hypocoagulability (<20 µm/min) at admission and in the following days.

Patients	Time	% of Patients with TDX-V			*n*
		>29 µm/min	20–29 µm/min	<20 µm/min	
Severely ill	At admission ^#^	75.6	17.3	7.1	241
	By day 2 ^##^	28.0	18.6	53.4	724
	Average in the next days ^##^	25.3 ± 4.4	14.6 ± 1.3	60.1 ± 5.4	2692
Mildly ill	At admission ^#^	79.6	14.3	6.1	1129
	By day 2 ^##^	25.6	32.3	42.1	726
	Average in the next days ^##^	26.8 ± 2.3	33.9 ± 5.3	39.3 ± 5.9	1917

^#^ Blood samples were collected before LMWHs were given. ^##^ Blood samples were collected at the trough level of pharmacological activity of LMWH.

**Table 4 jcm-14-01966-t004:** Cox regressions for lethal outcomes and thrombotic complications in severely ill patients.

Model	*n*	Variable	Base Model HR (95% CI), *p*-Value	Model Adjusted for Age, Sex and BMI HR (95% CI), *p*-Value
Death	393	Intense hypercoagulability (IH) ^#^	1.72 (1.38–2.14), ***	1.68 (1.35–2.10), ***
		Age	-	1.54 (1.32–1.80), ***
		Male	-	1.19 (0.90–1.57), ns
		BMI	-	1.20 (1.05–1.38), **
Thrombosis	318	Intense hypercoagulability (IH) ^#^	3.16 (2.29–4.37), ***	3.19 (2.31–4.41), ***
		Age	-	0.94 (0.71–1.25), ns
		Male	-	1.05 (0.62–1.77), ns
		BMI	-	0.94 (0.71–1.26), ns

^#^ The regression variable IH reflects the duration of intense hypercoagulability. IH = 0 if the patient experienced intense hypercoagulability for <25% of the time up to the event (death in the first model and thrombosis in the second model). IH = 1 if it was ≥25% and IH = 2 if it was ≥50% of the time up to the event. HR—hazard ratio, CI—confidence interval, BMI—body mass index, **—*p*-value < 0.01, ***—*p*-value < 0.001, ns—not significant.

**Table 5 jcm-14-01966-t005:** Comparison of anticoagulation guidelines with the study results.

Guideline	Mildly Ill Inpatients	Severely Ill Inpatients
ISTH	Recommend therapeutic anticoagulation in select patients.	Intermediate- or therapeutic-dose anticoagulation not recommended over prophylactic-dose heparin.
NIH COVID-19 Guideline	Recommend therapeutic-dose heparin for patients who have a D-dimer above the upper limit of normal, require low-flow oxygen, and have no increased bleeding risk.	Recommend prophylactic-dose heparin. Recommends against the use of intermediate-dose (e.g., enoxaparin 1 mg/kg daily) and therapeutic-dose anticoagulation for VTE prophylaxis, except in a clinical trial.
World Health Organization	Suggest standard thromboprophylaxis dosing.	Suggest standard thromboprophylaxis dosing.
Temporal Guidelines of the Ministry of Health of Russian Federation, v. 12	LMWH or UFH is recommended in at least prophylactic doses for all hospitalized patients until discharge. Dose escalation to intermediate or therapeutic levels may be considered for high D-dimer levels or additional thrombotic risk factors.	Routine escalation to intermediate/therapeutic doses in ICU patients does not improve outcomes. Therapeutic doses are used for confirmed thrombotic events.
Current Study Findings	Consistent with current guidelines. Escalated anticoagulation reduced the risk of hypercoagulability, thereby decreasing the likelihood of thrombotic events.	Both standard and escalated anticoagulation did not prevent intense hypercoagulability in some patients. TDX-V values exceeding 50 μm/min could serve as a clinical decision point for switching to UFH infusion with the rate adjusted using APTT or Thrombodynamics.

Table adapted from [12].

## Data Availability

The original data presented in the study are openly available in GitHub at https://github.com/shakhidzhanov-s/COVITRO/ (accessed on 10 March 2025).

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
