# Peer review of "Severely Ill COVID-19 Patients May Exhibit Hypercoagulability Despite Escalated Anticoagulation"

_jcm, 2025, doi:10.3390/jcm14061966_

Round 1

Reviewer 1 Report

Comments and Suggestions for Authors

I read with interest the manuscript submitted by Shakhidzhanov and colleagues on the impact, or rather lack of, increasing anticoagulation among patients with severe COVID-19 infections.

The manuscript fits the journal's scope, and is of interest to the readership of the journal. The manuscript presents a multicenter study, including 1654 patients with COVID-19. In this study, authors have found that, despite the increased dosing of anticoagulation therapy, patients still exhibited hypercoagulability, further contributing to their condition and possibly death.

While the study has its merits, e.g., in its design and sample size, the manuscript in its current form requires several critical modifications.

These modifications are as follows:

1- The introduction is poorly presented and is somewhat confusing to the reader. For example, the second sentence of the introduction stated "Clinical trials have shown that antiplatelet therapy does not prevent thrombosis and does not reduce mortality". and continued in the second paragraph to state that anticoagulation in COVID-19 patients did not reduce mortality or hypercoagulation events. Hence, the rationale behind conducting this study remains questionable at best.

2- Authors should clearly state their objectives, or hypothesis for their study in the last paragraph of the introduction to guide the reader through their manuscript.

3- Figures should be placed immediately after their first in-text reference. 

4- In the Figure legends, authors should clarify all the abbreviations, e.g., Ambul (figure 1), and UFH (figure 2).

5- Authors should not assume readers' awareness with the Russian Federation National Guidance for Treating COVID-19, and should briefly explain it in the Methods. 

6- What does "automatically translated" mean (in line 125)?

7- An essential aspect of the study design is the inclusion and exclusion criteria (which are missing), as well as clear definitions of the patients' classification (missing as well). Authors should clearly describe what they mean by Critically ill and Noncritically ill.

8- The term Critically and Noncritically ill are inaccurate and should be replaced with better and specific wording, e.g., Mild vs. Severe.

9- Authors should state their IRB ethical approval in a clear manner (issuing authority, approval number and date).

10- How did the authors conduct their statistical analysis? and what was the rationale for not checking the distribution of the data?

11- Authors should start their results section by describing their study cohort's characteristics, e.g., age, gender, BMI ..etc. This should precede the analytical results.  

12- Similarly, Table titles should be placed above the table, while all the abbreviations should be placed under the table.

13- The interpretation of P value, as presented in Table 1, is confusing. For example, P value of the first variable (Death), how did the authors compare 51.6 to 0?

14- Authors should remove the column "total" from Table 1 and present it separately, and the comparisons in a separate table.

15- In its current status, the discussion section is poor and underreferenced. Authors should refrain from dividing the discussion section into subsections (4.1 and 4.2 are pointless). 

16- Authors should be focused in their discussion. For example, authors should thoroughly examine their results against published studies (confirming/contradicting their results) and discuss such findings.

17- Hence, referring to results figures and tables in the discussion section is inappropriate. 

18- Another missing key aspect of the discussion was the comparison of the tools used in this study to other well-established tools used in COVID-19 that include parameters of the coagulation profile, e.g., CoMPred tool (10.3390/biomedicines11102649).

19- The limitation section is very vague. Authors should clearly state their limitations, why they think it is a limitation, and how it may have influenced their results.  

Author Response

RESPONSE TO REVIEWER 1

We thank you for reviewing our manuscript and for your valuable comments, suggestions and questions. The changes we made to the manuscript are highlighted with a cyan marker.

  1. The introduction is poorly presented and is somewhat confusing to the reader. For example, the second sentence of the introduction stated "Clinical trials have shown that antiplatelet therapy does not prevent thrombosis and does not reduce mortality". and continued in the second paragraph to state that anticoagulation in COVID-19 patients did not reduce mortality or hypercoagulation events. Hence, the rationale behind conducting this study remains questionable at best.

We recognize that the original introduction may not have been fully clear to the reader. To address this, we revised it to provide greater detail and logical coherence, while also more clearly articulating the objectives of our study.

In the updated introduction, we emphasize that SARS-CoV-2 infection profoundly disrupts hemostatic balance through a complex interplay of multiple concurrent mechanisms. These include platelet hyperactivation, elevated levels of von Willebrand factor, thromboinflammatory processes, and plasma coagulation cascade hypercoagulability driven by increased circulating extracellular vesicles containing tissue factor.

One of the effective strategies for reducing the risks of thrombotic complications and mortality has been the use of low-molecular-weight heparins (LMWH). However, it has become evident that LMWH dose escalation is insufficient to fully mitigate these risks. Such unmitigated risks may stem from other concurrent factors, such as hyperactive platelets or thromboinflammatory processes that remain unaffected by LMWH. Another potential concurrent factor is that escalated LMWH still do not effectively prevent plasma hypercoagulability in severely ill patients, especially at the trough level of the pharmacological activity and thus do not reduce the risks of thrombosis and mortality.

The objective of our study was to determine whether escalated (intermediate and therapeutic) LMWH prevented blood hypercoagulability in COVID-19 patients at the trough level of the pharmacological activity.

Please see the corresponding changes in lines 50-63, 78-90, 98-100.

  1. Authors should clearly state their objectives, or hypothesis for their study in the last paragraph of the introduction to guide the reader through their manuscript.

We stated our objective in the last paragraph of the introduction.

Please see the corresponding changes in lines 98-100.

  1. Figures should be placed immediately after their first in-text reference.

We placed all figures and tables immediately after their first in-text reference.

Please see the corresponding changes in lines 110, 183, 287, 314, 335, 389, 427, 450, 453.

  1. In the Figure legends, authors should clarify all the abbreviations, e.g., Ambul (figure 1), and UFH (figure 2).

We clarified all the abbreviations in the Figure legends.

Please see the corresponding changes in lines 123, 190-192, 341.

  1. Authors should not assume readers' awareness with the Russian Federation National Guidance for Treating COVID-19, and should briefly explain it in the Methods.

We added the following information:

“It states that all hospitalized COVID-19 patients should receive LMWH or fondaparinux sodium at least in prophylactic doses. UFH could be used as an alternative. Prophylaxis should be maintained during the whole hospitalization period. Patients with severe COVID-19 symptoms, additional thrombotic risk factors, or markedly elevated D-dimer levels were candidates for escalation to intermediate or therapeutic doses of LMWH/UFH. At the same time, randomized controlled trials have not demonstrated that routine escalation to intermediate or therapeutic doses of anticoagulants improved clinical outcomes in critically ill patients in the ICU. Therapeutic dosages of LMWH/UFH/fondaparinux sodium should have been indicated for a minimum of three months in patients with suspected/confirmed pulmonary embolism (PE) or deep vein thrombosis (DVT).”

Please see the corresponding change in lines 152-162.

  1. What does "automatically translated" mean (in line 125)?

We clarified this line: “version 12 translated using Google Translate© is attached”

Please see the corresponding change in line 151.

  1. An essential aspect of the study design is the inclusion and exclusion criteria (which are missing), as well as clear definitions of the patients' classification (missing as well). Authors should clearly describe what they mean by Critically ill and Noncritically ill.

We clarified this point.

Inclusion criteria: Inclusion criteria: age between 18 and 100 years, a requirement for hospitalization, consent from the patient or their representative to participate in the study, a preliminary diagnosis of COVID-19, and the patient's condition being classified as mild or severe. Exclusion criteria: refusal of the patient or their representative to participate in the study, COVID-19 diagnosis was not confirmed through PCR, pregnancy.

Patients were classified as severely ill if they required for admission to the intensive care unit (ICU) during hospitalization. Patients who did not require for admission to ICU during hospitalization were classified as mildly ill.

Please see the corresponding changes in lines 110-115, 124-126, 284-286.

  1. The term Critically and Noncritically ill are inaccurate and should be replaced with better and specific wording, e.g., Mild vs. Severe.

We replaced Critically ill with Severely ill and Noncritically ill with Mildly ill in all text and in the Title.

  1. Authors should state their IRB ethical approval in a clear manner (issuing authority, approval number and date).

We clarified this with the following line: “The study protocol was approved on June 30, 2020 by the Independent Ethical Committee of Dmitry Rogachev National Medical Research Center of Pediatric Hematology, Oncology, and Immunology, Ministry of Healthcare of Russia, Moscow (approval â„–: 4/2020, issued by Rumyantsev AG and Ataullakhanov FI)”.

Please see the corresponding changes in lines 249-253.

  1. How did the authors conduct their statistical analysis? and what was the rationale for not checking the distribution of the data?

The statistical analysis of continuous data was conducted using the two-sided Mann-Whitney U test with continuity correction. For categorical data, analysis was performed using the two-sided Fisher's exact test. We did not check the distributions for normality since it is not necessary for large samples and alters the conditional Type I error rates [Rochon J, Gondan M, Kieser M. To test or not to test: Preliminary assessment of normality when comparing two independent samples. BMC Med Res Methodol. 2012; 12:81.]. The analysis was performed and graphs were plotted using the R programming language.

Please see the corresponding changes in lines 258-260, 281-282.

  1. Authors should start their results section by describing their study cohort's characteristics, e.g., age, gender, BMI ..etc. This should precede the analytical results.

We added the following information in the beginning of Results:

“Severely ill patients were older (68 vs 62), had higher rate of VTE and arterial thromboembolism (ATE), more pronounced inflammatory state (CRP 98.8 vs 40.1 mg/l, white blood cell count 7.6 vs 5.8 x109/l, metabolic (glucose 7.0 vs 6.0 mmol/l) and coagulation (D-dimer 1256 vs 487 ng/mL) disturbances (Tab. 1). Severely ill patients also had higher rates of cardiovascular comorbidities, such as coronary artery disease, arterial hypertension, diabetes mellitus and chronic kidney disease. Severely ill patients also had higher computed tomography scores and required higher levels of respiratory support at admission. Among severely ill 25.3% received IL6/IL6R blockers, 2.4% – JAK inhibitors, 25.6% – steroids (Tab. 1).

The most common form of VTE was deep vein thrombosis, which occurred in 20.1% of severely ill patients and 2.4% of mildly ill patients. The most common form of ATE was limb artery thrombosis, which occurred in 1.9% of severely ill patients and 0.1% of mildly ill patients (Tab. 1).”

Please see the corresponding changes in lines 297-319.

  1. Similarly, Table titles should be placed above the table, while all the abbreviations should be placed under the table.

We placed all the abbreviations under the tables.

Please see the corresponding changes in lines 294-296, 463.

  1. The interpretation of P value, as presented in Table 1, is confusing. For example, P value of the first variable (Death), how did the authors compare 51.6 to 0?

We acknowledge that this point requires clarification. In such cases, we substituted 1 instead of 0 since this would serve as an upper estimate of the p-value. We applied this approach only if there were more than 10 events in the other group.

Please see the corresponding changes in table 1 (highlighted with cyan) and in lines 291-292, 296 (n.a.).

  1. Authors should remove the column "total" from Table 1 and present it separately, and the comparisons in a separate table.

We moved the column “total” into the supplementary material (Tab. S1).

Please see the corresponding changes in table 1 and table S1 in the supplementary material.

  1. In its current status, the discussion section is poor and underreferenced. Authors should refrain from dividing the discussion section into subsections (4.1 and 4.2 are pointless).

We acknowledge that the Discussion section in the previous text had these problems. To address this, we have added more references and provided a more detailed comparison of our results with existing literature, as you suggested in comment #16. We added the following sections in Discussion (references can be found in the Reference section of the manuscript):

“The characteristics of our study cohort were consistent with those reported in other studies: our severely ill patients had comorbidities, e.g. diabetes, hypertension, coronary artery disease, chronic kidney disease, which is typical [46]. Also, at admission, patients had elevated levels of D-dimer, fibrinogen, glucose, lactate dehydrogenase, and C-reactive protein, and these markers were notably higher in severely ill patients, which is also typical [47,48]. Prothrombin time was slightly elevated above the normal range, while APTT remained within the reference range, which is consistent with other study [49]. The overall incidence of venous and arterial thromboembolic complications in our study was comparable to previously published data [50-53].”

“A potential concurrent factor causing intense hypercoagulability can be a high level of factor VIII, which was observed in some patients with coronavirus infection [54,55]. The study [54] showed that patients with factor VIII levels >314 IU/dl, had 16.6-fold higher hazard rate for death. Intense hypercoagulability can also be caused by massive cell-release of procoagulant extracellular vesicles induced by severe inflammation [6,7].”

“Some studies [54,62] have suggested that in COVID-19 patients, an elevated risk of mortality may be indicated by prolonged APTT rather than shortened APTT. Pro-longed APTT may result from a deficiency in clotting factors due to consumptive coagulopathy, which is preceded by a hypercoagulable state. However, since this test is not well-suited for detecting hypercoagulability [61], APTT only becomes sensitive at advanced stages of the disease, when treatment may no longer be effective. Another potential cause of prolonged APTT could be antiphospholipid syndrome, which has been observed in some patients with COVID-19 [63].”

“In addition to the blood coagulation, platelets may play a significant role in the development of thrombotic complications during coronavirus infection. Their hyper-reactivity could drive persistent hypercoagulability in patients of our study. For instance, large platelet aggregates were detected in high numbers in the blood of patients [64]. In another study, using machine learning methods and single-cell transcriptomics, distinct platelet subpopulations that correlate with disease severity were identified [31]. In another study, a disruption in the gene expression profile of platelets was observed, along with increased platelet hyperactivity [32]. On the other hand, some studies [36,37] have shown that platelet activity in COVID-19 patients is actually sup-pressed, particularly in severely ill patients [36]. It is also worth noting that clinical trials of antiplatelet drugs did not demonstrate their effectiveness in reducing the risks of fatal outcomes and thrombosis [17,38,39]. However, it is also possible that platelets play an important role in the propagation of coronavirus infection [33]. In our study, we did not examine the role platelets, but we acknowledge that they could have played an important role in the development of thrombotic complications in some our patients.”

“The risk of a fatal outcome may increase in cases of coronavirus infection not only due to abnormalities in the plasma or platelet components of hemostasis, but also as a result of impaired immune system function [65]. Currently, there are numerous models for assessing the risk of severe illness and death in such patients, including those with severe illness [66-69]. In addition to standard coagulation test values, these models often include indicators such as the level of C-reactive protein, the neutrophil-to-lymphocyte ratio, and erythrocyte sedimentation rate.”

“In prior studies, Thrombodynamics, Thromboelastography assays showed that most patients were hospitalized with blood hypercoagulability, but it was not present at the peak pharmacological activity of LMWH [70]. In our study, we found that following the initiation of treatment and anticoagulant therapy, approximately 70% of patients experienced normalization of plasma coagulability, or in some cases, even developed hypocoagulability at the trough levels of LMWH pharmacological activity. These results align with those obtained from the Thrombin Generation assay (TG), which showed that COVID-19 patients have hypercoagulability at admission but standard thromboprophylaxis reduced TG to levels of healthy controls at the trough level of anticoagulation [71,72].”

“Alternative anticoagulation strategies, such as transitioning to UFH infusion, might be considered for patients with TDX-V >50 μm/min. Our study demonstrated that UFH infusion was associated with fewer instances of intense hyper-coagulability. Therefore, TDX-V values exceeding 50 μm/min could serve as a clinical decision point for switching to UFH infusion with the rate adjusted using APTT or Thrombodynamics [43] (Tab. 4).”

We also added a table with comparison of current anticoagulation guidelines with the study results – Table 4.

We added 24 new references in Discussion.

We removed subsection 4.1 and 4.2 from Discussion.

Please see the corresponding changes in lines 490-498, 531-543, 549-579, 587, 612-616, Table 4.

  1. Authors should be focused in their discussion. For example, authors should thoroughly examine their results against published studies (confirming/contradicting their results) and discuss such findings.

We took this suggestion into account while working on the previous suggestion.

  1. Hence, referring to results figures and tables in the discussion section is inappropriate.

We removed such references from Discussion.

  1. Another missing key aspect of the discussion was the comparison of the tools used in this study to other well-established tools used in COVID-19 that include parameters of the coagulation profile, e.g., CoMPred tool (10.3390/biomedicines11102649).

We added the following section in Discussion:

“The risk of a fatal outcome may increase in cases of coronavirus infection not only due to abnormalities in the plasma or platelet components of hemostasis, but also as a result of impaired immune system function [65]. Currently, there are numerous models for assessing the risk of severe illness and death in such patients, including those with severe illness [66-69]. In addition to standard coagulation test values, these models of-ten include indicators such as the level of C-reactive protein, the neutrophil-to-lymphocyte ratio, and erythrocyte sedimentation rate.”

Please see the corresponding changes in lines 564-570.

  1. The limitation section is very vague. Authors should clearly state their limitations, why they think it is a limitation, and how it may have influenced their results.

We acknowledge that this section was vague and rewrote it to make it clearer. The changes were made in the following paragraphs:

“One of the key limitations of our study is the absence of anti-factor Xa activity measurements, which are commonly used to assess the efficacy of LMWH therapy. The lack of Anti-Xa data leaves a gap in validating LMWH effectiveness in our research. Combining these measurements with Thrombodynamics could provide a more precise evaluation of hypercoagulability, helping to determine whether it results from insufficient pharmacological activity of the anticoagulant. On the other hand, Anti-factor Xa activity alone is insufficient for a comprehensive assessment of thrombotic complication risk [76, 77], underscoring the need to complement it with additional blood coagulation assays, such as Thrombodynamics. It is worth noting that Thrombodynamics demonstrates sensitivity to heparin comparable to that of anti-factor Xa activity [43].”

“In our study 8.2% of venous thrombotic events occurred when TDX-V showed plasma hypocoagulability. Currently, we cannot explain this. Possibly, these events could result from platelet hyperreactivity or thrombocytosis. Hence, platelet activity assays might be necessary to get a full picture of clotting state in COVID-19 patients. Lack of these data also limits out study.”

“We chose to analyze venous and arterial thromboses collectively, as TDX-V values were elevated prior to both types of events. Our results provide an average risk estimate for thrombotic events, encompassing both venous and arterial occurrences. However, it is important to note that the risks for venous thrombosis may differ from those for arterial thrombosis.”

“We were unable to obtain baseline TDX-V values in patients who received their first anticoagulation treatment in an ambulance. This practice was introduced midway through the study; however, by that time, we had already collected a substantial number of baseline TDX-V values, which we believe adequately reflected the baseline coagulability of the entire cohort. Additionally, a few patients had significant gaps between TDX-V measurements and were therefore excluded from TDX-V-related analyses. Nevertheless, these gaps were not associated with illness severity, and we do not believe they impacted the overall validity of our results.”

“The study extended over a year and included the beta, gamma, and delta variants of COVID-19, each of them could affect hemostasis differently. We were unable to determine the virus variants in our patients for comparison with our data and validation of our results for different virus variants. Also, whether these results are valid with the subsequent waves and the current continuous level of COVID-19 episodes is not known.”

Please see the corresponding changes in lines 620-629, 632-634, 638-655.

Reviewer 2 Report

Comments and Suggestions for Authors

This study presents important findings on hypercoagulability in critically ill COVID-19 patients, particularly introducing the concept of “intense hypercoagulability” (TDX-V ≥50 μm/min) that persists despite escalated anticoagulation. The research highlights the limitations of conventional anticoagulation strategies and the potential of Thrombodynamics (TDX) as a superior tool for detecting hypercoagulability compared to traditional coagulation assays like APTT. The findings suggest that personalized anticoagulation strategies may be needed, and that a combined monitoring approach using Thrombodynamics and D-dimer could improve risk stratification.

The study also emphasizes the dynamic nature of hypercoagulability in COVID-19 patients, with rapid changes in coagulation status observed. While the research provides a structured anticoagulation strategy for critically ill patients using LMWH and UFH, it also demonstrates that escalation of anticoagulation alone may be insufficient for certain high-risk individuals.

Major issues

  1. Anti-Xa activity is the standard clinical measurement for monitoring LMWH effectiveness, yet the study does not include any Anti-Xa activity measurements. Given that trough levels of LMWH pharmacological activity are a key aspect of the study, the absence of Anti-Xa data leaves a gap in the validation of LMWH efficacy. The authors should acknowledge this limitation and discuss how Anti-Xa monitoring might complement Thrombodynamics assessments.
  2. The study successfully identifies intense hypercoagulability but does not fully explore its biological mechanisms. Specifically, the potential role of platelet activation and aggregation in driving persistent hypercoagulability is not discussed in depth. The authors should consider citing and discussing relevant literature on platelet-driven hypercoagulability in COVID-19, including:

1) Nishikawa, M., Kanno, H., Zhou, Y., Xiao, T.H., Suzuki, T., Ibayashi, Y., Harmon, J., Takizawa, S., Hiramatsu, K., Nitta, N. and Kameyama, R., 2021. Massive image-based single-cell profiling reveals high levels of circulating platelet aggregates in patients with COVID-19. Nature communications, 12(1), p.7135.

2) Qiu, X., Nair, M.G., Jaroszewski, L. and Godzik, A., 2024. Deciphering Abnormal Platelet Subpopulations in COVID-19, Sepsis and Systemic Lupus Erythematosus through Machine Learning and Single-Cell Transcriptomics. International Journal of Molecular Sciences, 25(11), p.5941.

3) Manne, B.K., Denorme, F., Middleton, E.A., Portier, I., Rowley, J.W., Stubben, C., Petrey, A.C., Tolley, N.D., Guo, L., Cody, M. and Weyrich, A.S., 2020. Platelet gene expression and function in patients with COVID-19. Blood, The Journal of the American Society of Hematology, 136(11), pp.1317-1329.

  • Nishikawa et al. (2021, Nature Communications): Demonstrated high levels of circulating platelet aggregates in COVID-19 patients.
  • Qiu et al. (2024, IJMS): Used machine learning and single-cell transcriptomics to identify abnormal platelet subpopulations associated with fatal outcomes.
  • Manne et al. (2020, Blood): Investigated COVID-19-related platelet gene expression changes that could contribute to prothrombotic states.
  1. While the study suggests that increasing LMWH administration frequency rather than simply escalating the dose might be more effective, this remains a hypothesis and needs further validation.
  2. While the study identifies patients at risk of intense hypercoagulability, it does not establish a clear threshold for when alternative anticoagulation strategies like direct thrombin inhibitors should be considered, should TDX-V ≥50 μm/min be used as a clinical decision point for changing anticoagulation therapy? Please define actionable cutoffs would help translate these findings into practical clinical guidance.
  3. The study states that electronic medical records were used for data collection, but the dataset is not stored in a public repository. If sharing the full dataset is not possible due to ethical or institutional constraints, the authors should at least provide detailed metadata and analysis scripts.
  4. The study acknowledges that current guidelines (ISTH, NIH, WHO) recommend prophylactic anticoagulation for critically ill patients, but it does not provide a detailed comparison with these guideline. Please include direct comparison table summarizing how the study’s findings align or contradict existing guidelines would be valuable.

Minor Comments

  1. “As can be seen form Figure 3B, these intense hypercoagulability tails did not shift 299.” → “from” instead of “form”.
  2. “Data extraction was retrieval ensured and documented. 117” → This sentence is unclear and should be reworded.
  3. “Figure 1. flow of participants in the study.” → Use a clearer title for Figure 1.
  4. Figure 2 improvements:
  5. Add a color legend description.
  6. Use standard flowchart symbols (e.g., diamonds for decisions, rectangles for actions) to improve clarity.
  7. Include dosing information for the decision pathways.
  8. “Fig. 3AB). 279” → Should be “Fig. 3A,B).”.
  9. Inconsistencies in patient numbers:
  10. “Figure 3B. Total 3860 patients, why Ther. n = 3939?”
  11. “Figure S1I. Total 3860 patients, why n = 4449?”

Ensure patient numbers are consistent throughout the manuscript.

  1. Supplementary document “NationalGuidance.docx” has typos: “Intraalveolar Hemorrhage” should be “Intra-alveolar Hemorrhage”.

Author Response

RESPONSE TO REVIEWER 2

We thank you for reviewing our manuscript and for your valuable comments, suggestions and questions. The changes we made to the manuscript are highlighted with a cyan marker.

  1. Anti-Xa activity is the standard clinical measurement for monitoring LMWH effectiveness, yet the study does not include any Anti-Xa activity measurements. Given that trough levels of LMWH pharmacological activity are a key aspect of the study, the absence of Anti-Xa data leaves a gap in the validation of LMWH efficacy. The authors should acknowledge this limitation and discuss how Anti-Xa monitoring might complement Thrombodynamics assessments.

We acknowledge that the absence of Anti-Xa data leaves a gap in the validation of LMWH efficacy in our study. Anti-Xa measurements combined with Thrombodynamics could provide a more precise evaluation of hypercoagulability, helping to determine whether it results from insufficient pharmacological activity of the anticoagulant. However, Anti-factor Xa activity alone is insufficient for a comprehensive assessment of thrombotic complication risk [74, 75] (references can be found in the Reference section), underscoring the need to complement it with additional clotting assays, such as Thrombodynamics. Also, Thrombodynamics demonstrates sensitivity to heparin comparable to that of anti-factor Xa activity [43]. Hence, we decided not to perform this assay during the study.

Please see the corresponding changes in lines 620-629.

  1. The study successfully identifies intense hypercoagulability but does not fully explore its biological mechanisms. Specifically, the potential role of platelet activation and aggregation in driving persistent hypercoagulability is not discussed in depth. The authors should consider citing and discussing relevant literature on platelet-driven hypercoagulability in COVID-19, including:

1) Nishikawa, M., Kanno, H., Zhou, Y., Xiao, T.H., Suzuki, T., Ibayashi, Y., Harmon, J., Takizawa, S., Hiramatsu, K., Nitta, N. and Kameyama, R., 2021. Massive image-based single-cell profiling reveals high levels of circulating platelet aggregates in patients with COVID-19. Nature communications, 12(1), p.7135.

2) Qiu, X., Nair, M.G., Jaroszewski, L. and Godzik, A., 2024. Deciphering Abnormal Platelet Subpopulations in COVID-19, Sepsis and Systemic Lupus Erythematosus through Machine Learning and Single-Cell Transcriptomics. International Journal of Molecular Sciences, 25(11), p.5941.

3) Manne, B.K., Denorme, F., Middleton, E.A., Portier, I., Rowley, J.W., Stubben, C., Petrey, A.C., Tolley, N.D., Guo, L., Cody, M. and Weyrich, A.S., 2020. Platelet gene expression and function in patients with COVID-19. Blood, The Journal of the American Society of Hematology, 136(11), pp.1317-1329.

  • Nishikawa et al. (2021, Nature Communications): Demonstrated high levels of circulating platelet aggregates in COVID-19 patients.
  • Qiu et al. (2024, IJMS): Used machine learning and single-cell transcriptomics to identify abnormal platelet subpopulations associated with fatal outcomes.
  • Manne et al. (2020, Blood): Investigated COVID-19-related platelet gene expression changes that could contribute to prothrombotic states.

We agree that our text lacked a discussion of the possible causes of hypercoagulability. Among these, an increase in the level of extracellular vesicles, elevated factor VIII levels, and platelet hyperactivity may be contributing factors. We have added the following paragraphs to the text:

 “A potential concurrent factor causing intense hypercoagulability can be a high level of factor VIII, which was observed in some patients with coronavirus infection [54,55]. The study [54] showed that patients with factor VIII levels >314 IU/dl, had 16.6-fold higher hazard rate for death. Intense hypercoagulability can also be caused by massive cell-release of procoagulant extracellular vesicles induces by severe inflammation [6,7].”

“In addition to the blood coagulation, platelets may play a significant role in the development of thrombotic complications during coronavirus infection. Their hyper-reactivity could drive persistent hypercoagulability in patients of our study. For in-stance, large platelet aggregates were detected in high numbers in the blood of patients [65]. In another study, using machine learning methods and single-cell transcriptomics, distinct platelet subpopulations that correlate with disease severity were identified [31]. In another study, a disruption in the gene expression profile of platelets was observed, along with increased platelet hyperactivity [32]. On the other hand, some studies [36,37] have shown that platelet activity in COVID-19 patients is actually sup-pressed, particularly in severely ill patients [36]. It is also worth noting that clinical trials of antiplatelet drugs did not demonstrate their effectiveness in reducing the risks of fatal outcomes and thrombosis [17,38,39]. However, it is also possible that platelets play an important role in the propagation of coronavirus infection [33]. In our study, we did not examine the role platelets, but we acknowledge that they could have played an important role in the development of thrombotic complications in some our patients.”

Please see the corresponding changes in lines 539-543, 549-563.

  1. While the study suggests that increasing LMWH administration frequency rather than simply escalating the dose might be more effective, this remains a hypothesis and needs further validation.

We acknowledge this. We modified the following line “It is possible that more frequent administration of LMWH without daily-dose escalation could be more effective in prevention of intense hypercoagulability, however, this hypothesis needs further validation.”

Please see the corresponding changes in lines 609-611.

  1. While the study identifies patients at risk of intense hypercoagulability, it does not establish a clear threshold for when alternative anticoagulation strategies like direct thrombin inhibitors should be considered, should TDX-V ≥50 μm/min be used as a clinical decision point for changing anticoagulation therapy? Please define actionable cutoffs would help translate these findings into practical clinical guidance.

Thank you for this valuable suggestion. In our study we showed that patients receiving UFH infusion less frequently have intense hypercoagulability. Hence, we can suggest switching to UFH infusion with the rate adjusted using APTT or Thrombodynamics when TDX-V >50 um/min.

Regarding direct thrombin and factor Xa inhibitors – at the time of the study, we were aware that in certain circumstances, these inhibitors do not adequately address hypercoagulability, which can explain high thrombosis frequency when they are used [Ataullakhanov FI, Dashkevich NM, Ovsepyan RA, et al. Heparin and Direct Oral Anticoagulants have Different Effects on the Phases of Activation and Spatial Spread of Blood Coagulation. Thromb Haemost. Online ahead of print]. This is because these inhibitors are reversible and, upon dissociation from their target, they fail to suppress hypercoagulability. Heparins, due to their irreversible inhibition of activated coagulation factors, are better suited for managing hypercoagulable states. Therefore, we would not recommend using them in this situation.

Please see the corresponding changes in lines 612-616.

  1. The study states that electronic medical records were used for data collection, but the dataset is not stored in a public repository. If sharing the full dataset is not possible due to ethical or institutional constraints, the authors should at least provide detailed metadata and analysis scripts.

We upload all permitted information and scripts on https://github.com/shakhidzhanov-s/COVITRO/ . It allows to get figures 3, 4B, 5, 6, S4, S5, S6.

Please see the corresponding changes in line 256.

  1. The study acknowledges that current guidelines (ISTH, NIH, WHO) recommend prophylactic anticoagulation for critically ill patients, but it does not provide a detailed comparison with these guideline. Please include direct comparison table summarizing how the study’s findings align or contradict existing guidelines would be valuable.

Thank you for this valuable suggestion. We included such comparison in the Discussion section – table 4.

  1. “As can be seen form Figure 3B, these intense hypercoagulability tails did not shift 299.” → “from” instead of “form”.

Thank you.

Please see the corresponding changes in line 363.

  1. “Data extraction was retrieval ensured and documented. 117” → This sentence is unclear and should be reworded.

Thank you, we reworded it “Data extraction was performed with documentation of the retrieval process.”

Please see the corresponding changes in lines 147-148.

  1. “Figure 1. flow of participants in the study.” → Use a clearer title for Figure 1.

We made a new title “Distribution of hospitalized patients by admission route, hospitals, and severity of illness.”

Please see the corresponding changes in line 119.

  1. Figure 2 improvements:
  • Add a color legend description.

We did not notice that it was missing at the time of manuscript submission. Currently, it appears as follows: “Figure 2. Algorithm for the use of LMWH and UFH in the treatment of COVID-19 in adult patients in an inpatient setting. Anticoagulation de-escalation was recommended for patients with kidney injury. It could also be temporarily lowered or discontinued in cases of bleeding, intubation, or surgical intervention. A table below the flowchart shows doses of heparins, used during the study. LMWH – low-molecular-weight heparin, UFH – unfractionated heparin, DVT – deep vein thrombosis, PE – pulmonary embolism, s/c – subcutaneously, TID – three time a day, BID – two times a day.”

Please see the corresponding changes in lines 186-192.

  • Use standard flowchart symbols (e.g., diamonds for decisions, rectangles for actions) to improve clarity.

We made a new flowchart.

Please see figure 2.

  • Include dosing information for the decision pathways.

We included such information below the flowchart.

Please see figure 2.

  1. “Fig. 3AB). 279” → Should be “Fig. 3A,B).”.

We corrected this error.

Please see the corresponding changes in line 337

  1. Inconsistencies in patient numbers:
  • “Figure 3B. Total 3860 patients, why Ther. n = 3939?”

We have added clarification to this figure. This number represents the total count of measurements, which is higher than the number of patients because multiple clot growth rate measurements could be taken for a single patient during hospitalization. We have clarified this in the caption for Figure 3:

“(A) – prophylactic: 458 patients with a total of 1035 measurements, intermediate: 558 patients with a total of 1526 measurements, therapeutic: 259 patients with a total of 593 measurements. (B) – prophylactic: 144 patients with a total of 327 measurements, intermediate: 452 patients with a total of 1281 measurements, therapeutic: 882 patients with a total of 3939 measurements. (C) - Prophylactic: 42 patients with a total of 79 measurements, intermediate: 122 patients with a total of 352 measurements, therapeutic: 245 patients with a total of 1151 measurements.”

Please see the corresponding changes in lines 350-357.

  • “Figure S1I. Total 3860 patients, why n = 4449?”

The reason is similar. We have clarified this in the caption for Figure S1: “A number of patients in each group and a total number of measurements are shown in the legends.”

Please see the corresponding changes in Figure’s S1 legend.

  1. Supplementary document “NationalGuidance.docx” has typos: “Intraalveolar Hemorrhage” should be “Intra-alveolar Hemorrhage”.

We corrected this error. Unfortunately, we were unable to catch all the typos that occurred during the translation using Google Translate.

Round 2

Reviewer 1 Report

Comments and Suggestions for Authors

I congratulate the authors for their efforts in addressing most of the concerns and comments raised in the first round of review. The manuscript is undoubtedly in a much better state than the original submission. However, a couple of issues remain and require the authors attention:

1) Comment No. 11 in the first round of revision remains unaddressed. The authors should start their results by clearly describing their cohort and include Table S1 as their first table, descriptive, not comparative as the current one.

2) In the authors' response to comment No.18, the authors should finish their statement with appropriate referencing "“The risk of a fatal outcome may increase in cases of coronavirus infection not only due to abnormalities in the plasma or platelet components of hemostasis, but also as a result of impaired immune system function [65]. Currently, there are numerous models for assessing the risk of severe illness and death in such patients, including those with severe illness [66-69]. In addition to standard coagulation test values, these models of-ten include indicators such as the level of C-reactive protein, the neutrophil-to-lymphocyte ratio, and erythrocyte sedimentation rate (10.3390/medicina60040602)". 

Reviewer 2 Report

Comments and Suggestions for Authors

The authors have revised the manuscript accordingly. 

Author Response

We thank you for reviewing our manuscript.